# On Generalization Bounds for Projective Clustering

**Maria Sofia Bucarelli**[1]    **Matilde Fjeldsø Larsen**[2] **Chris Schwiegelshohn**[2] **Mads Bech Toftrup**[2]

[1]Department of Computer, Control and Management Engineering Antonio Ruberti,
Sapienza University of Rome, Italy
[2]Department of Computer Science, Aarhus University, Denmark

`mariasofia.bucarelli@uniroma1.it`    `201805091@post.au.dk`
`{schwiegelshohn,toftrup}@cs.au.dk`

## Abstract

Given a set of points, clustering consists of finding a partition of a point set into $k$ clusters such that the center to which a point is assigned is as close as possible. Most commonly, centers are points themselves, which leads to the famous $k$-median and $k$-means objectives. One may also choose centers to be $j$ dimensional subspaces, which gives rise to subspace clustering. In this paper, we consider learning bounds for these problems. That is, given a set of $n$ samples $P$ drawn independently from some unknown, but fixed distribution $\mathcal{D}$, how quickly does a solution computed on $P$ converge to the optimal clustering of $\mathcal{D}$? We give several near optimal results. In particular,

1. For center-based objectives, we show a convergence rate of $\tilde{O}\left(\sqrt{k/n}\right)$. This matches the known optimal bounds of [Fefferman, Mitter, and Narayanan, Journal of the Mathematical Society 2016] and [Bartlett, Linder, and Lugosi, IEEE Trans. Inf. Theory 1998] for $k$-means and extends it to other important objectives such as $k$-median.
2. For subspace clustering with $j$-dimensional subspaces, we show a convergence rate of $\tilde{O}\left(\sqrt{kj^2/n}\right)$. These are the first provable bounds for most of these problems. For the specific case of projective clustering, which generalizes $k$-means, we show a convergence rate of $\Omega\left(\sqrt{kj/n}\right)$ is necessary, thereby proving that the bounds from [Fefferman, Mitter, and Narayanan, Journal of the Mathematical Society 2016] are essentially optimal.

## 1 Introduction

Among the central questions in machine learning is, given a sample of $n$ points $P$ drawn from some unknown but fixed distribution $\mathcal{D}$, how well does a classifier trained on $P$ generalize to $\mathcal{D}$? The probably most popular way to formalize this question is, given a loss function $L$ and optimal solutions $\mathcal{S}_P$ and $\mathcal{S}_\mathcal{D}$ for sample $P$ and distribution $\mathcal{D}$, respectively, how the empirical excess risk $L(\mathcal{D}, \mathcal{S}_P) - L(\mathcal{D}, \mathcal{S}_\mathcal{D})$ decreases as a function of $n$. This paper focuses on loss functions associated with the clustering problem. Popular examples include $(k, z)$ clustering, which asks for a set of $k$ centers $\mathcal{S} \subset \mathbb{R}^d$ minimizing the cost $\sum_{p\in P} \min_{s\in\mathcal{S}} \|p - s\|_2^z$ and more generally, $(k, j, z)$ subspace clustering which asks for a set of $k$ subspaces $\mathcal{U} := \{U_1, U_2, \ldots U_k\}$ minimizing $\sum_{p\in P} \min_{U_i\in\mathcal{U}} \|(I - U_i U_i^T)p\|_2^z$. Special cases include $(k, 1)$ clustering, known as $k$-median, $(k, 2)$ clustering known as $k$-means and $(k, j, 2)$ clustering known as projective clustering. Generally, there seems to be an interest in varying $z$, as letting $z$ tend towards 1 tends to result in outlier-robust clusterings. The problem is less widely explored for $z > 2$, although in particular for subspace

37th Conference on Neural Information Processing Systems (NeurIPS 2023).

approximation there is some recent work [27, 34, 79, 77]. Higher powers give more emphasis on outliers. For example, centralised moments with respect to the three and four norms are skewness and kurtosis, respectively, and are extensively employed in statistics, see Cohen-Addad et al. [31] for previous work on clustering with these measures. Fitting a mixture model with respect to skewness minimizes asymmetry around the target center. Studing the problems for $z \to \infty$ is very well motivated, as the $(1, \infty)$ clustering is equivalent to the minimum enclosing ball problem. Unfortunately, one often requires additional assumptions, as the minimum enclosing ball problem suffers from the curse of dimensionality [2], is very prone to outliers [25, 38].

Despite a huge interest and a substantial amount of research, so far optimal risk bounds $\tilde{O}\left(\sqrt{k/n}\right)$[1] for the $k$-means problem have been established, see the seminal paper by Fefferman et al. [39] for the upper bound and Bartlett et al. [10] for nearly matching lower bounds. For general $(k, z)$-clustering problems, the best known results prove a risk bound of $O\left(\sqrt{kd/n}\right)$ [10]. For $(k, j, 2)$ clustering, the best known bounds of $\tilde{O}\left(\sqrt{kj/n}\right)$ are due to Fefferman et al. [39]. Thus, the following question naturally arises:

> Is it possible to obtain optimal generalization bounds for all $(k, j, z)$-clustering objectives?

We answer this question in the affirmative whenever $j$ and $z$ are constant, which seems to be the most relevant case in practise [76]. Specifically, we show

- The excess risk bound for $(k, z)$-clustering when given $n$ independent samples from an unknown fixed distribution $\mathcal{D}$ is bounded by $\tilde{O}\left(\sqrt{k/n}\right)$, matching the lower bound of [10].

- The excess risk bound for $(k, j, z)$-clustering when given $n$ independent samples from an unknown fixed distribution $\mathcal{D}$ is bounded by $\tilde{O}\left(\sqrt{kj^2/n}\right)$.

- There exists a distribution such that the excess risk for the $(k, j, 2)$-clustering problem is at least $\Omega\left(\sqrt{kj/n}\right)$, matching the upper bound of Fefferman et al. [39] up to polylog factors.

We note that we assume $z$ to be a constant, which is the case for projective clustering, $k$-means and $k$-median. For non-constant $z$, the dependency on $z$ is exponential.

## 1.1 Related work

The most basic question one could answer is if the empirical estimation performed on $P$ is consistent, i.e. as $n \to \infty$, whether the excess risk tends to $0$. This was shown in a series of works by Pollard [64, 66], see also Abaya and Wise [1]. Subsequent work then analyzed the convergence rate of the risk. The first works in this direction proved convergence rates of the order $\tilde{O}(1/\sqrt{n})$ without giving dependencies on other parameters [22, 65]. Linder et al. [55] gave an upper bound of $O(d^{3/2}\sqrt{k/n})$. Linder [54] improved the upper bound to $O(d\sqrt{k/n})$. Bartlett et al. [8] showed an upper bound $O(\sqrt{kd/n})$ and gave a lower bound of $\Omega(\sqrt{k^{1-4/d}/n})$. Motivated by applications of clustering for high dimensional kernel spaces [7, 18, 20, 21, 37, 39, 56, 58, 67, 78, 80, 81], research subsequently turned its efforts towards minimizing the dependency on the dimension. Biau et al. [14] presented an upper bound of $O(k/\sqrt{n})$, see also the work by Clémençon [26]. Fefferman et al. [39] gave a matching upper bound of the order $O(\sqrt{k/n})$, which was later recovered by using techniques from Foster and Rakhlin [45] and Liu [57]. Further improvements require additional assumptions of the distribution $\mathcal{D}$, see Antos et al. [5], Levrard [52], Li and Liu [53]. For subspace clustering, there have only been results published for the case $z = 2$ [39, 50, 72], for which the state of the art provides a $\tilde{O}\left(\sqrt{kj/n}\right)$ risk bound due to Fefferman et al. [39]. A highly related line of research originated with the study of coresets for compression. For Euclidean $(k, z)$ clustering, coresets with space bounds of $\tilde{O}\left(k/\varepsilon^{2+z}\right)$ have been established [30, 32], which roughly corresponds to a error rate of

---

[1]$\tilde{O}$ hides logarithmic terms, i.e. we consider $O\left(\sqrt{k/n} \cdot \mathrm{polylog}(k, n)\right) = \tilde{O}\left(\sqrt{k/n}\right)$.

$\tilde{O}\left(\sqrt[2+z]{k/n}\right)$ as a function of the size of the compression. For the specific case of $k$-median and $k$-means, coresets with space bounds of $\tilde{O}\left(k^{(2z+2)/(z+2)}/\varepsilon^2\right)$ are known [33], which corresponds to a error rate of $\tilde{O}\left(\sqrt{k^{(2z+2)/(z+2)}/n}\right)$. Both results are optimal for certain ranges of $\varepsilon$ and $k$ [47] and while these bounds are worse than what we hope to achieve for generalization, many of the techniques such as terminal embeddings are relevant for both fields. For $(k, j, z)$ clustering, coresets are only known to exist under certain assumptions, where the provable size is $\tilde{O}\left(\exp(k, j, \varepsilon^{-1})\right)$ [40, 44].

## 2    Preliminaries

We use $\|x\|_p := \sqrt[p]{\sum |x_i|^p}$ to denote the $\ell_p$ norm of a vector $x$. For $p \to \infty$, we define the limiting norm $\|x\|_\infty = \max x_i$. Further, we refer to the $d$-dimensional unit Euclidean ball by $B_2^d$, i.e. $x \in B_2^d$ is a vector in $\mathbb{R}^d$ and $\|x\|_2 := \sqrt{\sum_{i=1}^d x_i^2} \leq 1$. Let $U$ be a $d \times j$ orthogonal matrix, i.e., with pairwise and orthogonal columns with unit Euclidean norm. We say that $UU^T$ is the projection matrix associated with $U$. Let $z$ be a positive integer. Given any set $\mathcal{S}$ of $k$ points in $B_2^d$ we denote the $(k, z)$-clustering cost for a point set $P$ with respect to solution $\mathcal{S}$ as

$$\text{cost}(P, \mathcal{S}) := \sum_{p \in P} \min_{s \in \mathcal{S}} \|p - s\|_2^z.$$

Special cases include $k$-means ($z = 2$) and $k$-median ($z = 1$). Similarly, given a collection $\mathcal{U}$ of $k$ orthogonal matrices of rank at most $j$, we denote the $(k, j, z)$-clustering cost of a point set $P$ as

$$\text{cost}(P, \mathcal{U}) := \sum_{p \in P} \min_{U \in \mathcal{U}} \|(I - UU^T)p\|_2^z.$$

The specific case $(k, j, 2)$ is often known as projective clustering in literature. The cost vector $v^{\mathcal{S}, P} \in \mathbb{R}^{|P|}$, respectively $v^{\mathcal{U}, P} \in \mathbb{R}^{|P|}$ has entries $v_p^{\mathcal{S}} = \min_{s \in \mathcal{S}} \|p - s\|_2^z$, respectively $v_p^{\mathcal{U}} = \min_{U \in \mathcal{U}} \|(I - UU^T)p\|_2^z$ for $p \in P$. We will omit $P$ from $v^{\mathcal{S}, P}$ and $v^{\mathcal{U}, P}$, if $P$ is clear from context. The overall cost is $\|v^{\mathcal{S}}\|_1 = \sum_{p \in P} \min_{s \in \mathcal{S}} \|p - s\|_2^z$ and $\|v^{\mathcal{U}}\|_1 = \sum_{p \in P} \min_{U \in \mathcal{U}} \|(I - UU^T)p\|_2^z$. The set of all cost vectors is denoted by $V$.

Let $\mathcal{D}$ be an unknown but fixed distribution on $B_2^d$ with probability density function $\mathbb{P}$. For any solution $\mathcal{S}$, respectively $\mathcal{U}$, we define $\text{cost}(\mathcal{D}, \mathcal{S}) := \int_{p \in B_2^d} \min_{s \in \mathcal{S}} \|p - s\|^z \cdot \mathbb{P}[p]dp$ and $OPT := \min_{\mathcal{S}} \text{cost}(\mathcal{D}, \mathcal{S})$ and respectively $\text{cost}(\mathcal{D}, \mathcal{U}) := \int_{p \in B_2^d} \min_{U \in \mathcal{U}} \|(I - UU^T)p\|^z \cdot \mathbb{P}[p]dp$ and $OPT := \min_{\mathcal{U}} \text{cost}(\mathcal{D}, \mathcal{U})$. Let $P$ be a set of $n$ points sampled independently from $\mathcal{D}$. We denote the cost of the empirical risk minimizer on $P$ by $OPT_P := \frac{1}{n} \min_{\mathcal{S}} \|v^{\mathcal{S}}\|_1$, and respectively, $OPT_P := \frac{1}{n} \min_{\mathcal{U}} \|v^{\mathcal{U}}\|_1$. The excess risk of $P$ with respect to a set of cost vectors is denoted by

$$\mathcal{E}_{|P|}(V) := \mathbb{E}_P[OPT_P] - OPT.$$

Finally, we use the notion of a net. Let $(V, dist)$ be a metric space, $\mathcal{N}(V, dist, \varepsilon)$ is an $\varepsilon$-net of the set of vectors $V$, if for all $v \in V \; \exists \; v' \in \mathcal{N}(V, dist, \varepsilon)$ such that $dist(v, v') \leq \varepsilon$. We will particularly focus on nets for cost vectors induced by $(k, z)$-clustering and $(k, j, z)$-clustering defined as follows, prior work has proposed similar nets for coresets and sublinear algorithms for $(k, z)$ clustering [31].

**Definition 2.1** (Clustering Nets). A set $\mathcal{N}_\varepsilon$ of $|P|$-dimensional vectors is an $\varepsilon$-clustering net if for every cost vector $v$ obtained from a solution $\mathcal{S}$ or $\mathcal{U}$, there exists a vector $v' \in \mathcal{N}_\varepsilon$ with $\|v' - v\|_\infty \leq \varepsilon$

A slightly weaker condition as required by these nets requiring only $\|v' - v\|_2 \leq \varepsilon\sqrt{n}$ would also be sufficient. Nevertheless, we are not able to show better bounds when relaxing the condition and having a point-wise guarantee may be of independent interest.

## 3    Outline and technical contribution

Due to space restrictions, the full proofs are provided in the supplementary material. In this section, we endeavour to present a complete and accessible overview of the key ideas behind the theorems.

Let $P$ be a set of $n$ points sampled independently from some unknown but fixed distribution $\mathcal{D}$. To show that the excessive risk with respect to clustering objectives is in $\tilde{O}(f(n))$ for some function $f$, it is sufficient to show two things. First, that for the optimal solution $\mathcal{U}_{\text{OPT}}$, the clustering cost estimated using $P$ is close to the true cost. Second, any solution that is more expensive than $\mathcal{U}_{\text{OPT}}$ does not become too cheap when evaluated on $P$. Both conditions are satisfied if for any solution $\mathcal{U}$

$$\left| \frac{1}{n}\text{cost}(P,\mathcal{U}) - \text{cost}(\mathcal{D},\mathcal{U}) \right| \in \tilde{O}(f(n)).$$

Showing $\left| \frac{1}{n}\text{cost}(P,\mathcal{U}_{\text{OPT}}) - \text{cost}(\mathcal{D},\mathcal{U}_{\text{OPT}}) \right| \in O(\sqrt{1/n})$ with good probability is typically a straightforward application of concentration bounds such as Chernoff's bound. In fact, these concentration bounds show something even stronger. Given $t$ solutions $\mathcal{U}_1,\ldots \mathcal{U}_t$, we have

$$\mathbb{E}_P \sup_{\mathcal{U}_i} \left| \frac{1}{n}\text{cost}(P,\mathcal{U}_i) - \text{cost}(\mathcal{D},\mathcal{U}_i) \right| \in O\left( \sqrt{\frac{\log t}{n}} \right). \tag{1}$$

What remains is to bound the number of solutions $t$.

**Clustering nets and dimension reduction for center based clustering**    Unfortunately, the total number of expensive clusterings in Euclidean space is infinite, making a straightforward application of 1 useless. Nets as per Definition 2.1 are now typically used to reduce the infinite number of solutions to a finite number. Specifically, one has to show that by preserving the costs of all solutions in the net, the cost of any other solution is also preserved. Using basic techniques from high dimensional computational geometry, it is readily possible to prove that a $\varepsilon$-net for $(k,j,z)$ clustering of size $\exp(k \cdot j \cdot d \cdot \log \varepsilon^{-1})$ exists, where $d$ is the dimension of the ambient space. Plugging this into Equation 1 and setting $\varepsilon^{-1} = n^2$ then yields a generalization bound of the order $O\left( \sqrt{kjd \log n/n} \right)$. Unfortunately, this leads to a dependency on $d$, which is suboptimal. To improve the upper bounds, we take inspiration from coreset research. For $(k,z)$-clustering, a number of works have investigated dimension reduction techniques known as terminal embeddings, see [11, 46]. Given a set of points $P \in \mathbb{R}^d$, a terminal embedding $f : \mathbb{R}^d \to \mathbb{R}^m$ guarantees $\|p - q\|_2 = (1 \pm \varepsilon) \cdot \|f(p) - f(q)\|_2$ for any $p \in P$ and $q \in \mathbb{R}^d$. Terminal embeddings are very closely related to the Johnson-Lindenstrauss lemma, see [16, 28, 60] for applications to clustering, but more powerful in key regard: only one of the points is required to be in $P$. The added guarantee extended to arbitrary $q \in \mathbb{R}^d$ due to terminal embeddings allows us to capture all possible solutions. There are also even simpler proofs for $k$-mean that avoid this machinery entirely, see [39, 45, 57]. Unfortunately, these arguments are heavily reliant on properties of inner products and are difficult to extend to other values of $z$. The terminal embedding technique may be readily adapted to $(k,z)$-clustering, though some care in the analysis must be made to avoid the worse dependencies on the sample size necessitated for the corset guarantee, described as follows.

**Improving the union bound via chaining:**    To illustrate the chaining technique, consider the simple application of the union bound for a terminal embedding with target dimension $m = \Theta(\varepsilon^{-2} \log n)$, see the main result of Narayanan and Nelson [62]. Replacing the dependency on $d$ with an appropriately chosen parameters and plugging the resulting net $N_\varepsilon$ of size $\exp(k\varepsilon^{-2} \log n \log \varepsilon^{-1})$ yields a generalization bound of $O\left( \sqrt[4]{k \log^2 n/n} \right)$ for $(k,z)$ clustering. We improve on this using a chaining analysis, see [30, 32] for its application to coresets for $(k,z)$ clustering and [39] for $(k,j,2)$ clusterings. Specifically, we use a nested sequence of nets $N_{1/2}, N_{1/4}, N_{1/8}, \ldots, N_{2^{-2\log n}}$. Note that for every solution $\mathcal{S}$, we may now write $\text{cost}(p,\mathcal{S})$ for any $p \in P$ as a telescoping sum

$$\text{cost}(p,\mathcal{S}) = \sum_{h=0}^{\infty} \text{cost}(p,\mathcal{S}_{2^{-(h+1)}}) - \text{cost}(p,\mathcal{S}_{2^{-h}})$$

with $,\mathcal{S}_{2^{-h}} \in N_h$ and $\text{cost}(p,\mathcal{S}_1)$ being set to 0. We use this as follows. Suppose for some solution $\mathcal{S}$, we have solutions $\mathcal{S}_{2^{-h}} \in N_{2^{-h}}$ and $\mathcal{S}_{2^{-(h+1)}} \in N_{2^{-(h+1)}}$. Then $|\text{cost}(p,\mathcal{S}_{2^{-h}}) - \text{cost}(p,\mathcal{S}_{2^{-(h+1)}})| \leq O(2^{-h}) |\text{cost}(p,\mathcal{S}_{2^{-h}}) - \text{cost}(p,\mathcal{S})|$ for all $p \in P$. Instead of applying the union bound for a small set of solutions, we apply the union bound along every pair of

solutions appearing in the telescoping sum. Using arguments similar to Equation 1, we then obtain

$$\mathbb{E}_P \sup_{\substack{\mathcal{S}_{2^{-h}} \times \mathcal{S}_{2^{-(h+1)}} \\ \in N_h \times N_{h+1}}} \left| \frac{1}{n}\mathrm{cost}(P, \mathcal{S}_{2^{-h}}) - \frac{1}{n}\mathrm{cost}(P, \mathcal{S}_{2^{-(h+1)}}) \right|$$

$$= 2^{-h} \cdot \tilde{O}\left( \sqrt{\frac{\log(|N_h| \cdot |N_{h+1}|)}{n}} \right) = 2^{-h} \cdot \tilde{O}\left( \sqrt{\frac{k \cdot 2^{2h} \cdot \mathrm{polylog}(k/2^h)}{n}} \right) \in \tilde{O}\left( \sqrt{\frac{k}{n}} \right)$$

This is the desired risk bound for $(k, z)$ clustering. To complete the argument in a rigorous fashion, we must now merely combine the decomposition of $\mathrm{cost}(P, \mathcal{S})$ into the telescoping sum with the learning rate that we just derived. Indeed, this already provides a simple way of obtaining a bound on the risk of the order $\tilde{O}\left( \sqrt{k/n} \right)$, which turns out to be optimal. In summary, to apply the chaining technique successfully, the following two properties are sufficient: (i) the dependency on $\varepsilon$ in the net size can be at most $\exp(\tilde{O}(\varepsilon^{-2}))$, as the increase in net size is then met with a corresponding decrease between successive estimates along the chain and (ii) the nets have to preserve the cost up to an additive $\varepsilon$ for *every* sample point $p$. The second property is captured by Definition 2.1. Both properties impose restrictions on the dimension reductions that can be successfully integrated into the chaining analysis.

**Dimension reduction for projective clustering:** It turns out that extending this analysis $(k, j, z)$ clustering is a major obstacle. While the chaining method itself uses no particular properties of $(k, z)$ clustering, the terminal embeddings needed to obtain nets cannot be applied to subspaces. Indeed, terminal embeddings by the very nature of their guarantee, cannot be linear[2], and hence a linear structure such as a subspace will not be preserved. At this stage, there are a number of initially promising candidates that can provide alternative dimension reduction methods. For example, the classic Johnson-Lindenstrauss lemma can be realized via a random embedding matrix and, moreover, preserves subspaces, see for example [69, 23, 28]. Unfortunately, as remarked by [46], there is an inherent difficulty in applying Johnson-Lindenstrauss type embeddings even for $(k, z)$ clustering coresets and the same arguments also apply for generalization bounds.

An alternative dimension reduction method based on principal component analysis was initially proposed by [44] for $(k, j, 2)$, see also [28] and most notably [74] for a different variant that applies to arbitrary $(k, j, z)$ objectives. For $(k, j, 2)$ clustering, it states that a dimension reduction on the first $O(D/\varepsilon)$ principal components preserves the projective cost of all subspaces of dimension $D$. Since $(k, j, 2)$ clustering is a special case of a $k \cdot j$ dimensional projection, it implies that $O(kj/\varepsilon)$ dimensions are sufficient. Given that these dimension reductions are based on PCA-type methods, they are linear and therefore seem promising initially. Unfortunately, this technique has serious drawbacks. It does not satisfy the requirements for Definition 2.1, only preserving the cost on aggregate rather then per individual point, and thus cannot be combined with the chaining technique[3]. Without the chaining technique, the best bound one can hope for is of the order $\tilde{O}\left( \sqrt[3]{k^2 j^2/n} \right)$, which falls short of what we are aiming for.

Another important technique used to quantify optimal solutions of $(k, j, z)$ clustering initially proposed by [73] and subsequently explored by [43, 35] and has frequently seen use in coreset literature [40, 46]. Succinctly, it states that a $(1 + \varepsilon)$ approximate solution to the $(1, j, z)$ clustering problem of a point set $P$ is contained in a subspace spanned by $\tilde{O}(j^2/\varepsilon)$ input points of $P$. While this result improves over PCA for large values of $k$, applying it only yields a learning rate of the order $O(\sqrt[3]{kj^3/n})$. It turns out that this technique has the exact same limitations as PCA, namely that costs per point are not preserved, and thus only offers a different tradeoff in parameters.

**Our new insight:** Given the state of the art, designing a dimension reduction technique that would enable the application of the chaining technique might seem hopeless, and indeed, we were not able to prove such. The key insight that allows us to bypass these bottlenecks is to find a dimension reduction

---

[2]Consider an embedding matrix $S \in \mathbb{R}^{d \times m}$. Clearly, there exists some vector $x \in \mathbb{R}^d$ that is in the kernel of $S$ whenever $m < d$, hence for any vector $p$, $\|p - (x + p)\|_2$ cannot be preserved.

[3]PCA as well as the other potential alternative dimension reduction techniques also do not satisfy the relaxed definition that would be sufficient for the analysis to go through.

that applies not to all solutions $\mathcal{U}$, but only to a certain subset of them. Indeed, we show that for any point set $P$ contained in the unit ball and any subspace $\mathcal{U}$ of dimension $j$, there exists a subspace $S$ spanned by $O(j/\varepsilon^2)$ points of $P$ such that for every point $p$: $|\text{cost}(p, \mathcal{U}) - \text{cost}(p_S, \mathcal{U}_S)| \leq \varepsilon$. This is similar to the guarantee provided by [73] but stronger in that it (i) applies to arbitrary subspaces, which is required for the chaining analysis, and (ii) applies to each point of $P$ individually, rather than for the entire point set $P$ on aggregate. We then augment the chaining analysis by applying a union bound over all $\binom{|P|}{j/\varepsilon^2}$ possible dimension reductions, thereby capturing all solutions $\mathcal{U}$. We are unaware of any previously successful attempts at integrating multiple dimension reductions within a chaining analysis and believe that the technique may be of independent interest.

# 4 Useful results from learning theory

Our goal is to bound the rate with which the empirical risk decreases for clustering problems. For a fixed set of $n$ points $P$ and a set of functions $F : P \to \mathbb{R}$, we define the Rademacher complexity ($Rad_n$) and the Gaussian complexity ($G_n$) wrt $F$ respectively as

$$Rad_n(F) = \frac{1}{n} \cdot \mathbb{E}_r \sup_{f \in F} \sum_{p \in P} f(p) \cdot r_p \qquad G_n(F) = \frac{1}{n} \cdot \mathbb{E}_g \sup_{f \in F} \sum_{p \in P} f(p) \cdot g_p$$

where $r_p$ are independent random variables following the Rademacher distribution, whereas $g_p$ are independent Gaussian random variables. In our case, we can think of $f$ as being associated to a solution $\mathcal{S}$ (respectively a solution $\mathcal{U}$) and $f(p) = \text{cost}(p, \mathcal{S}) = \min_{s \in \mathcal{S}} \|p - s\|_2^z$ (respectively $f(p) = \text{cost}(p, \mathcal{U}) = \min_{U \in \mathcal{U}} \|(I - UU^T)p\|_2^z$). Since we associate every $f$ with a cost vector $v^{\mathcal{S}}$, we will use $Rad_n(F)$ and $Rad_n(V)$ as well as $G_n(F)$ and $G_n(V)$ interchangeably. The following theorem is due to Bartlett and Mendelson. [9].

**Theorem 4.1** (Simplified variant of Theorem 8 of Bartlett and Mendelson [9])**.** *Consider a loss function $L : A \to [0, 1]$. Let $F$ be a class of functions mapping from $X$ to $A$ and let $(X_i)_{i=1}^n$ be independent samples from $\mathcal{D}$. Then, for any integer $n$ and any $\delta > 0$, with probability at least $1 - \delta$ over samples of length $n$, denoting by $\hat{\mathbb{E}}_n$ the empirical risk, every $f \in F$ satisfies*

$$\mathbb{E}L(f(X)) \leq \hat{\mathbb{E}}_n L(f(X)) + Rad_n(F) + \sqrt{\frac{8 \ln 2/\delta}{n}}.$$

Thus, in order to bound the excess risk, Theorem 4.1 shows that it is sufficient to bound the Rademacher complexity. It is well known (see, for example, B.3 of Rudra and Wootters [68]) that $Rad_n(V) \leq \sqrt{2\pi} G_n(V)$. Thus we can alternatively bound the Gaussian complexity, which is sometimes more convenient. Note that if $V$ is the set of all cost vectors, clustering nets are mere $\mathcal{N}(V, \|.\|_\infty, \varepsilon)$. Using these nets, we can bound the Rademacher and Gaussian complexity. Indeed the following lemma holds.

**Lemma 4.2.** *Let $\mathcal{D}$ be a distribution over $B_2^d$ and let $P$ a set of $n$ points sampled from $\mathcal{D}$. Suppose that for a set of $n$-dimensional vector $V$, we have an absolute constant $C, \gamma > 0$ such that $\log |\mathcal{N}(V, \|.\|_\infty, \varepsilon)| \in O(\varepsilon^{-2} \log^\gamma(n\varepsilon^{-1})C)$. Then*

$$G_n(V) \in O\left(\sqrt{\frac{C \log^{\gamma+2} n}{n}}\right).$$

The specific types of nets used in our study and the size bounds for those nets will be the key to obtaining the desired upper bounds and will be detailed in the next section.

# 5 Generalization bounds for center-based clustering and subspace clustering

We start by giving our generalization bounds for center based clustering and subspace clustering problems. For subspace clustering problems, we first state the result for general $(k, j, z)$ clustering. An improvement for the special case $z = 2$ will be given later.

**Theorem 5.1.** *Let $\mathcal{D}$ be a distribution over $B_2^d$ and let $P$ be a set of $n$ points sampled from $\mathcal{D}$. For any set of $k$ points $\mathcal{S} \subset B_2^d$, we denote by $v^{\mathcal{S}}$ the $n$-dimensional cost vector of $P$ in solution $\mathcal{S}$*

with respect to the $(k, z)$-clustering objective. Moreover we denote by $v^{\mathcal{U}}$ the $n$-dimensional cost vector of $P$ in solution $\mathcal{U}$ with respect to the $(k, j, z)$-clustering objective. Let $V_z$ be the union of all cost vectors of $P$ for the center-based clustering and $V_{j,z}$ the union of all cost vectors for subspace clustering. Then with probability at least $1 - \delta$

$$\mathcal{E}_n(V_z) \in O\left(\sqrt{\frac{k \cdot \log^4 n}{n}} + \sqrt{\frac{\log 1/\delta}{n}}\right) \tag{2}$$

$$\mathcal{E}_n(V_{j,z}) \in O\left(\sqrt{\frac{k \cdot j^2 \cdot \log jn \cdot log^3 n}{n}} + \sqrt{\frac{\log 1/\delta}{n}}\right). \tag{3}$$

Following Theorem 4.1, it is sufficient to bound the Rademacher complexity in order to bound the excess risk. The Rademacher complexity is, up to lower order terms, equal to the Gaussian complexity, which, following Lemma 4.2 may be bounded by obtaining small nets with respect to the $\|.\|_\infty$ norm. We believe that the results on the bounds of the nets, may be of independent interest and we'll state these results in the following Lemma.

**Lemma 5.2.** *Let $\mathcal{D}$ be a distribution over $B_2^d$ and let $P$ a set of $n$ points sampled from $\mathcal{D}$, let $V_z$ be defined as in Theorem 5.1 let $V_{j,z}$ be defined as in Theorem 5.1. Then*

$$|\mathcal{N}(V_z, \|.\|_\infty, \varepsilon)| \leq \exp(O(1)z^3 \cdot k \cdot \varepsilon^{-2} \log n \cdot (\log(z) + \log(\varepsilon^{-1}))) \tag{4}$$

$$|\mathcal{N}(V_{j,z}, \|.\|_\infty, \varepsilon)| \leq \exp(O(1)(3z)^{z+2} \cdot k \cdot j \cdot \varepsilon^{-2}(\log n + j \log(j\varepsilon^{-1})) \log \varepsilon^{-1}). \tag{5}$$

Combining Lemma 5.2 with Lemma 4.2 now yields the immediate bound on the Rademacher and Gaussian complexity. Following the discussion from Section 3, we use terminal embeddings to prove the part of Lemma 5.2 pertaining to $(k, z)$ clustering, see Appendix B. Unfortunately, the terminal embedding technique is not admissible for obtaining nets for subspace clustering as clarified in Section 3. Thus, we use an entirely different approach. We show the existence of a collection of dimension reducing maps with subspace preserving properties. Fortunately, the number of dimension reducing maps is small. Our desired net sizes then follow by enumerating over all of these dimension reducing maps, and for the candidate solutions covered by each such dimension reducing map, we can find an efficient net. First, we introduce a slightly different, but closely related notion to $(1, j, z)$-nets.

**Definition 5.3** (Projective Nets). *Let $P \subset B_2^d$ be a set of points, and let $z$ be a positive integer. For a $d \times j$ matrix $S$ with columns that have at most unit norm and any point $p \in P$, define the projective cost as $\text{cost}_{proj}(p, S) = \|S^T p\|_2$. Let $V$ be the set of all projective cost vectors induced by such matrix $S$. We call a $\mathcal{N}(V, \|.\|_\infty, \varepsilon)$ a $(\varepsilon, j)$-projective net of $P$.*

On a high level, the proof largely relies on the following decomposition. Let $U$ be a candidate subspace and let $\Pi$ be a projection matrix used to approximate $\|(I - UU^T)p\|_2^z$ We have

$$\|(I-UU^T)p\|^2 = \underbrace{\|\Pi p\|^2}_{(1)} - \underbrace{\|U^T\Pi p\|^2}_{(2)} + \underbrace{\|(I-\Pi)p\|^2}_{(3)} - \underbrace{\|UU^T(I-\Pi)p\|^2}_{(4)} + \underbrace{2p^T\Pi UU^T(I-\Pi)p}_{(5)} \tag{6}$$

Here, we wish to select $\Pi$ such that $\|U^T(I - \Pi)p\|_2$ is small for all $p \in P$. Note that this implies that the terms $2p^T\Pi UU^T(I - \Pi)p$ and $\|UU^T(I - \Pi)p\|^2$ are small. For the term (2), we merely have to show that projective nets exist. If the number of $\Pi$ is small, we can further construct good nets for the terms (1) and (3) . We start by giving a bound for the projective nets. Our first Lemma 5.4 shows that if the points lie in a sufficiently low-dimensional space, such a net can be obtained by constructing a net $\mathcal{N}(B_2^d, \|.\|_2, \varepsilon')$ for a sufficiently small $\varepsilon'$.

**Lemma 5.4.** *Let $P \subset B_2^d$ be a set of points, and $z$ be a positive integer. Then there exists an $(\varepsilon, j)$-projective net of size $|\mathcal{N}(V, \|.\|_\infty, \varepsilon)| \leq \exp(O(1) \cdot d \cdot j \cdot \log(j\varepsilon^{-1}))$.*

To reduce the dependency on the dimension, we now use the following lemma. Essentially, it shows that in order to retain the properties of $U$, we can find a projection matrix $\Pi$ of rank at most $O(j\varepsilon^{-2})$.

**Lemma 5.5.** *Let $P \subseteq B_2^d$. For any orthogonal matrix $U \in \mathbb{R}^{j \times d}$, there exists $M \subseteq P$, with $|M| \in O(j \cdot \varepsilon^{-2})$, such that $\forall p \in P, \|U^T(I - \Pi_M)p\| \leq \varepsilon \cdot \|(I - \Pi_M)p\|$.*

We now use this lemma as follows. We can efficiently enumerate over all candidate $\Pi$, as Lemma 5.5 guarantees us that we only have to consider $\binom{n}{j \cdot \varepsilon^{-2}} \leq \exp(j \cdot \varepsilon^{-2} \log n)$ many different $M$ inducing projection matrices. This immediately gives us 0-nets for the terms (1) and (3). For each $\Pi$, we then apply Lemma 5.4, which gives us a net for term (2). Finally, by choice of $\Pi$, we can show that terms (4) and (5) are negligible.

## 5.1 Tight generalization bounds for projective clustering

For the specific case of $(k, j, 2)$ clustering, also known as projective clustering, we obtain an even better dependency on $j$. A similar bound can likely also be derived using the seminal work of [39], though the dependencies on $\log n$ and $\log 1/\delta$ are slightly weaker. The proof uses the main result by [45], itself heavily inspired by [39], and arguments related to bounding the Rademacher complexity of linear function classes. Crucially, it avoids the issue of obtaining an explicit dimension reduction entirely, but the approach cannot be extended to general $(k, j, z)$ clustering.

**Theorem 5.6.** *Let $\mathcal{D}$ be a distribution over $B_2^d$ and let $P$ a set of $n$ points sampled from $\mathcal{D}$. For any set $\mathcal{U}$ of $k$ orthogonal matrices of rank at most $j$, we denote by $v^{\mathcal{U}}$ the $n$-dimensional cost vector of $P$ in solution $\mathcal{U}$ with respect to the $(k, j, 2)$-clustering objective, i.e. $v_p^{\mathcal{U}} = \min_{U \in \mathcal{U}} \|(I - UU^T)p\|^2$. Let $V_{j,2}$ be the union of all cost vectors of $P$. Then with probability at least $1 - \delta$ for any $\gamma > 0$*

$$\mathcal{E}_n(V_{j,2}) \in O\left(\sqrt{\frac{kj}{n} \cdot \log^{3+\gamma}\left(\frac{n}{j}\right)} + \sqrt{\frac{\log 1/\delta}{n}}\right).$$

Finally, we also show that the bounds from Theorem 5.6 and [39] are optimal up to polylogarithmic factors.

**Theorem 5.7.** *There exists a distribution $\mathcal{D}$ supported on $B_2^d$ such that $\mathcal{E}_n(V_{j,2}) \in \Omega\left(\sqrt{(kj)/n}\right)$.*

The rough idea is to define a distribution $\mathcal{D}$ supported on the nodes of a $2kj$-dimensional simplex with some points having more probability mass and some points having smaller mass. Using the tightness of Chernoff bounds, we may then show that the probability of fitting a subspace clustering to a good fraction of the lower mass points is always sufficiently large.

## 6 Experiments

Theoretical guarantees are often notoriously conservative compared to what is seen in practice. In this section, we present empirical findings detailing whether the risk bounds from the previous sections are also the risk bounds one can expect when dealing with real datasets. Indeed, for the related question of computing coresets, experimetal work by [71] seems to indicate that the worst case bounds by [47] are not what one has to expect in practise for center based clustering. Generally, two properties can determine the risk decrease. First, the clusters may be well separated [4, 29]. Indeed, making assumptions to this end, there is also some theoretical evidence that a rate of $O(k/n)$ is possible [5, 52]. The other, somewhat related explanation is that if the ground truth consists of $k' < k$ clusters [13, 63], the dependency on $k$ will point more towards the smaller, true number of clusters. We run the experiments both for center based clustering, as well as subspace clustering. While the focus of the paper is arguably more on subspace clustering, the experiments are important in both cases. Although both problems are hard to optimize exactly, center based clustering is significantly more tractable and thus may lend better insight into practical learning rates. For example, we have an abundance of approximation algorithms for $(k, z)$ clustering [6, 61] whereas, even in the case of $(k, 1, z)$ clustering in two dimensions [49] it is not possible to find any finite approximation in polynomial time.

In the main body, we focus on $(k, 1, z)$ clustering, as there already exists a phase transition in terms of the computational complexity between the normal $k$-median and $k$-means problems and the $(k, 1, 1)$ and $(k, 1, 2)$ clustering objectives, while $j = 1$ still admits more positive results than other subspace clustering problems Agarwal et al. [3], Feldman et al. [41, 42].

**Datasets** We use four publicly available real-world datasets: Mushroom [70], Skin-Nonskin [12], MNIST [51], and Covtype [15]. Below, we show the results on the Covtype dataset, and the remaining

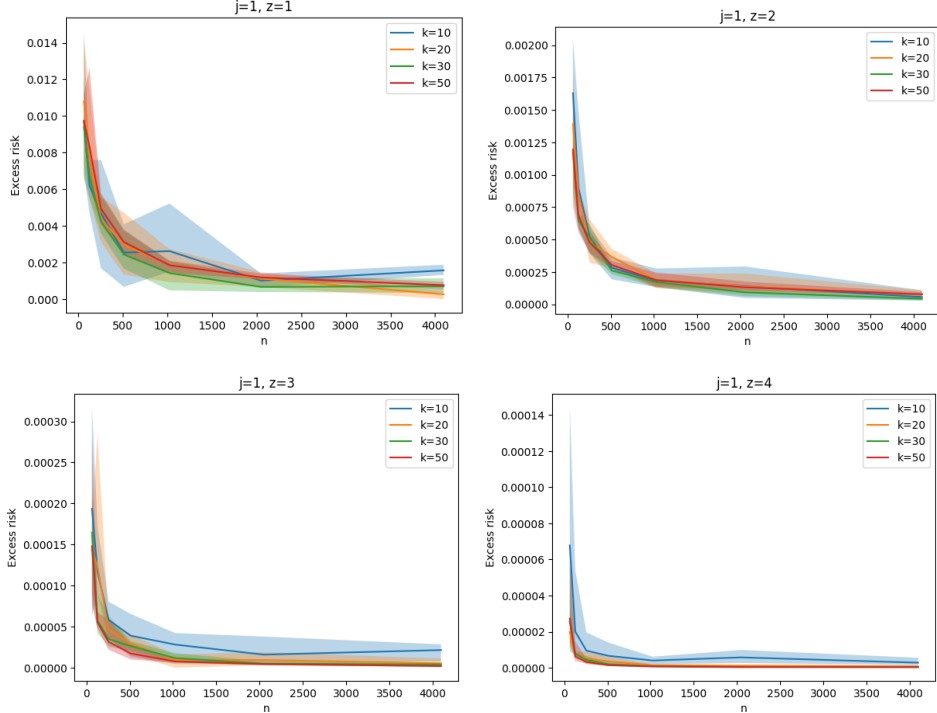

Figure 1: Excess risk for line clustering on Covtyp. Shaded areas show max-min intervals.

experiments are deferred to the supplementary material. Each dataset was normalized by the diameter, ensuring that all points lie in $B_2^d$.

Table 1: Datasets used for the experiments

| Dataset | Points | Dim | Labels |
|---|---|---|---|
| Mushrooms | 8,124 | 112 | 2 |
| MNIST | 60,000 | 784 | 10 |
| Skin_Nonskin | 245,057 | 3 | 2 |
| Covtype | 581,012 | 54 | 7 |

**Problem parameters and algorithms**  For both center based clustering as well as subspace clustering, we focus on the powers $z \in \{1, 2, 3, 4\}$. $z = 2$ is arguably the most popular and also the most tractable variant. $z = 1$ is the objective with the least susceptibility to outliers. Finally, we consider the cases $z = 3$, due to it minimizing asymmetry and $z = 4$ as a tractable alternative to the coverage objective $z \to \infty$. The excess risk is evaluated for $k \in \{10, 20, 30, 50\}$ for both center based and subspace clustering. Expectation maximization (EM) type algorithms are used for both center-based and subspace clustering, though this is a severe computational challenge fo $(1, j, z)$ clustering, if $z \neq 2$, see [24, 36]. Given a solution $\mathcal{S}$ we first assign every point to its closest center and subsequently recompute the center. For more details on initialization and concrete implementations, we refer to the supplementary material.

**Experimental setup and results**  To estimate the optimal cost $OPT$ for the two objective functions, we run the corresponding appropriate algorithms mentioned above ten times on the entire dataset $P$ and use the minimal objective value as an estimate for $OPT$. We obtain a sample $S_i$ of size $n$ by sampling uniformly at random and estimate the optimal cost for that sample, $OPT_i$. We repeat this 5 times. The empirical excess risk is calculated as $\mathcal{E}_n = \frac{1}{|P|} \sum_{i=1}^{5} \frac{\text{cost}(P, OPT_i)}{5} - OPT$. The excess risk for center-based clustering is evaluated on exponential-sized subset sizes $n \in \{2^6, 2^7, \ldots, 2^{12}\}$.

We fit a line of the form $c \cdot \frac{k^{q_1}}{n^{q_2}}$ where $c, q_1, q_2$ are the optimizeable parameters. Let $y_i$ be the excess risk in run $i$. Let $k_i$ and $n_i$ be the values of $k$ and $n$ in run $i$ and let $r$ be the total number of times the excess risk was evaluated for each combination of algorithm and dataset. We use gradient descent on the following loss to optimize the parameters $LSE = \sum_{i=1}^{r} \left( y_i - c \cdot \frac{k^{q_1}}{n^{q_2}} \right)^2$.

The results in Figure 1 show that the excess risk for subspace clustering decreases quicker for higher values of $z$, and we see a similar pattern for center-based clustering. The supplementary material contains more details on the empirical evaluations of center-based clustering. The best-fit lines shown in Tables 2 and 3 in the supplementary material indicate that the empirical excess risk values decrease slightly quicker than predicated by theory. The expected values are $q_1 = q_2 = 0.5$ and we observe $q_1, q_2$ around $0.44, 0.52$ respectively. For $k$ this indicates a slightly favorable dependency in practice. For $q_2$, we consider the difference to the theoretical bound of $0.5$ negligible. The choice of $z$ does not seem to have a significant impact on either finding. For subspace clustering, the dependency on $k$ is a bit more pronounced and increases slightly towards the theoretical guarantees. Contrary to hopes that margin or stability conditions might occur on practical datasets, the results indicate that the theoretical guarantees of the learning rate are near-optimal even in practice. Moreover, the rates were not particularly affected by either the choice of $z$ or by the dimension $j$ when analyzing subspace clustering.

## 7    Conclusion and open problems

In this paper, we presented several new generalization bounds for clustering objectives such as $k$-median and subspace clustering. When the centers are points or constant dimensional subspaces, our upper bounds are optimal up to logarithmic terms. For projective clustering, we give a lower bound showing that the results obtained by [39] are nearly optimal. A key novel technique was using an ensemble of dimension reduction methods with very strong guarantees.

An immediate open question is to which degree ensembles of dimension reductions can improve learning rates over a single dimension reduction. Is it possible to find natural problems where there is a separation between the embeddability and the learnablity of a class of problems, or given the ensemble, is it always possible to find a single dimension reduction with the guarantees of the ensemble? Another open question is motivated by the recent treatment of clustering through the lens of computational social choice [19]. Using current techniques from coresets [17] and learning theory [45], it seems difficult to improve over the learning rate of $O\left(\sqrt{k^2/n}\right)$ for the fair clustering problem specifically. It it possible to match the bounds for unconstrained clustering?

## Disclosure of Funding Acknowledgements

Maria Sofia Bucarelli was partially supported by projects FAIR (PE0000013) and SERICS (PE00000014) under the MUR National Recovery and Resilience Plan funded by the European Union - NextGenerationEU. Supported also by the ERC Advanced Grant 788893 AMDROMA, EC H2020RIA project "SoBigData++" (871042), PNRR MUR project IR0000013-SoBigData.it.

Chris Schwiegelshohn was supported by the Independent Research Fund Denmark (DFF) under a Sapere Aude Research Leader grant No 1051-00106B.

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

# Supplementary Materials

## A    Proof of Lemma 4.2

In this section, we include the proof of Lemma 4.2 and some preliminary facts that will be useful for the proof.

Let $r$ be a Rademacher vector, i.e. every entry $r_i$ is sampled independently uniformly from $\{-1, 1\}$. Further, we say that $g$ is a Gaussian vector if every entry $g_i$ is a standard Gaussian with mean 0 and variance 1. We have the following useful properties of Gaussians.

*Fact* A.1 (Appendix B.1 by [68]). Let $g_1, \ldots g_n$ be Gaussians with means $\mu_i$ and variances $\sigma_i^2$.

- If $\sigma_i^2 \leq \sigma^2$ for all $i$, then $\mathbb{E}[\max_{g_i} |g_i|] \leq 2\sigma\sqrt{2 \log n}$.

- If the Gaussians are independent, then $\sum_{i=1}^{n} a_i g_i$ is Gaussian distributed with mean $\sum_{i=1}^{n} a_i \mu_i$ and variance $\sum_{i=1}^{n} a_i^2 \sigma_i^2$.

- If the $g_i$ are independent standard Gaussians with mean 0 and variance 1, then $Y := \sum_{i=1}^{n} g_i^2$ is Chi-squared distributed with mean $\mathbb{E}[\sqrt{Y}] \in O(\sqrt{n})$.

Another result we need is the following

**Lemma A.2** (Lemma 5.2 of [75]). $|\mathcal{N}(B_2^d, \|.\|_2, \varepsilon)| \leq (1 + 2/\varepsilon)^d$.

We are now ready to prove the Lemma 4.2. The proof of Lemma is similar to arguments used to prove Dudley's theorem. We also write here the statement of the Lemma for the sake of completeness

**Lemma** (Lemma 4.2). *Let $\mathcal{D}$ be a distribution over $B_2^d$ and let $P$ be a set of $n$ points sampled from $\mathcal{D}$. Suppose that for a set of $n$-dimensional vectors $V$, we have absolute constants $C, \gamma > 0$ such that*

$$\log |\mathcal{N}(V, \|.\|_\infty, \varepsilon)| \in O(\varepsilon^{-2} \log^\gamma (n \cdot \varepsilon^{-1}) \cdot C). \tag{7}$$

*Then*

$$G_n(V) \in O\left(\sqrt{\frac{C \log^{\gamma+2} n}{n}}\right).$$

*Proof.* For ease of notation, we use solutions $\mathcal{S}$ induced by points, but the proof carries over without any modifications other than changing the notation to collections of subspaces $\mathcal{U}$.

Consider an arbitrary cost vector $v^{\mathcal{S}}$. We write $v^{\mathcal{S}}$ as a telescoping sum

$$v^{\mathcal{S}} := \sum_{h=0}^{\infty} v^{h+1, \mathcal{S}} - v^{h, \mathcal{S}}$$

where $v^0 = 0$ and $v^{i, \mathcal{S}}$ is a vector from $\mathcal{N}(V, \|.\|_\infty, 2^{-i})$ approximating $v^{\mathcal{S}}$. Observe that

$$\|v^{h+1, \mathcal{S}} - v^{h, \mathcal{S}}\|_\infty \leq \|v^{h+1, \mathcal{S}} - v^{\mathcal{S}} + v^{\mathcal{S}} - v^{h, \mathcal{S}}\|_\infty \leq 2 \cdot 2^{-h} \tag{8}$$

due to the triangle inequality. Thus we have

$$
\begin{aligned}
n \cdot G_n(V) &= \mathbb{E}_{P,g}\left[\sup_{\mathcal{S}}(v^{\mathcal{S}})^T g\right] = \mathbb{E}_{P,g}\left[\sup_{\mathcal{S}}\sum_{h=0}^{\infty}(v^{h+1,\mathcal{S}} - v^{h,\mathcal{S}})^T g\right] \\
&\leq \mathbb{E}_{P,g}\sum_{h=0}^{\infty}\left[\sup_{\mathcal{S}}(v^{h+1,\mathcal{S}} - v^{h,\mathcal{S}})^T g\right] \\
&= \mathbb{E}_{P,g}\sum_{h=0}^{\infty}\left[\sup_{\substack{v^{h+1,\mathcal{S}},v^{h,\mathcal{S}}\in \\ \mathcal{N}(V,\|.\|_\infty,2^{-(h+1)})\times\mathcal{N}(V,\|.\|_\infty,2^{-h})}}(v^{h+1,\mathcal{S}} - v^{h,\mathcal{S}})^T g\right] \\
&= \mathbb{E}_{P,g}\sum_{h=0}^{\log n}\left[\sup_{\substack{v^{h+1,\mathcal{S}},v^{h,\mathcal{S}}\in \\ \mathcal{N}(V,\|.\|_\infty,2^{-(h+1)})\times\mathcal{N}(V,\|.\|_\infty,2^{-h})}}(v^{h+1,\mathcal{S}} - v^{h,\mathcal{S}})^T g\right] \\
&\quad+\mathbb{E}_{P,g}\sum_{h=\log n}^{\infty}\left[\sup_{\substack{v^{h+1,\mathcal{S}},v^{h,\mathcal{S}}\in \\ \mathcal{N}(V,\|.\|_\infty,2^{-(h+1)})\times\mathcal{N}(V,\|.\|_\infty,2^{-h})}}(v^{h+1,\mathcal{S}} - v^{h,\mathcal{S}})^T g\right] \\
&= \mathbb{E}_{P,g}\sum_{h=0}^{\log n}\left[\sup_{\substack{v^{h+1,\mathcal{S}},v^{h,\mathcal{S}}\in \\ \mathcal{N}(V,\|.\|_\infty,2^{-(h+1)})\times\mathcal{N}(V,\|.\|_\infty,2^{-h})}}(v^{h+1,\mathcal{S}} - v^{h,\mathcal{S}})^T g\right] \qquad (9) \\
&\quad+\mathbb{E}_{P,g}\left[\sup_{\mathcal{S}}(v^{\mathcal{S}} - v^{\log n,\mathcal{S}})^T g\right] \qquad\qquad\qquad\qquad\qquad (10)
\end{aligned}
$$

We bound the terms 9 and 10 differently, starting with the latter.

For every $\mathcal{S}$

$$
(v^{\mathcal{S}} - v^{\log n,\mathcal{S}})^T g \leq \|v^{\mathcal{S}} - v^{\log n,\mathcal{S}}\|_2 \cdot \mathbb{E}[\|g\|_2],
$$

due to the Cauchy Schwarz inequality. Further,

$$
\|v^{\mathcal{S}} - v^{\log n,\mathcal{S}}\|_2 \leq \sqrt{n} \cdot \|v^{\mathcal{S}} - v^{\log n,\mathcal{S}}\|_\infty \leq \sqrt{n} \cdot 2^{-\log n} = \sqrt{\frac{1}{n}},
$$

which, combined with the third item in Fact A.1 yields

$$
\mathbb{E}_{P,g}\left[\sup_{\mathcal{S}}(v^{\mathcal{S}} - v^{\log n,\mathcal{S}})^T g\right] \in O\left(\sqrt{\frac{1}{n}} \cdot \sqrt{n}\right) = O(1). \qquad (11)
$$

We now consider the term 9. Due to the second item of Fact A.1, $(v^{h+1,\mathcal{S}} - v^{h,\mathcal{S}})^T g$ is Gaussian distributed with mean 0 and variance

$$
\sum_{i=1}^{n}(v^{h+1,\mathcal{S}} - v^{h,\mathcal{S}})_i^2 \leq 4n \cdot 2^{-2h}.
$$

Thus, we have, using the first item in Fact A.1

$$
\begin{aligned}
&\mathbb{E}_{P,g}\sum_{h=0}^{\log n}\left[\sup_{\substack{v^{h+1,\mathcal{S}},v^{h,\mathcal{S}}\in \\ \mathcal{N}(V,\|.\|_\infty,2^{-(h+1)})\times\mathcal{N}(V,\|.\|_\infty,2^{-h})}}(v^{h+1,\mathcal{S}} - v^{h,\mathcal{S}})^T g\right] \\
&\leq \sum_{h=0}^{\log n}\sqrt{32n \cdot 2^{-2h}\log\left|\mathcal{N}(V,\|.\|_\infty,2^{-(h+1)}) \times \mathcal{N}(V,\|.\|_\infty,2^{-h})\right|}
\end{aligned}
$$

$$
(12)
$$

Now using equation (7) we obtain that,

$$32n \cdot 2^{-2h} \log \left| \mathcal{N}(V, \|.\|_\infty, 2^{-(h+1)}) \times \mathcal{N}(V, \|\dot{\|}_\infty, 2^{-h}) \right| \in O(n \cdot \log^\gamma n)$$

So we have that

$$\sum_{h=0}^{\log n} \sqrt{32n \cdot 2^{-2h} \log \left| \mathcal{N}(V, \|.\|_\infty, 2^{-(h+1)}) \times \mathcal{N}(V, \|.\|_\infty, 2^{-h}) \right|} \in O(\sqrt{n \cdot \log^{\gamma+2} n}) \quad (13)$$

Adding the bounds (13) and (11) for Terms (10) and (9), respectively yields the claim. $\qquad \square$

Finally, we will frequently use the following triangle inequality extended to powers.

**Lemma A.3** (Triangle Inequality for Powers (Lemma A.1 of [60])). *Let $a, b, c$ be an arbitrary set of points in a metric space with distance function $d$ and let $z$ be a positive integer. Then for any $\varepsilon > 0$*

$$d(a,b)^z \le (1+\varepsilon)^{z-1} d(a,c)^z + \left( \frac{1+\varepsilon}{\varepsilon} \right)^{z-1} d(b,c)^z$$

$$|d(a,b)^z - d(a,c)^z| \le \varepsilon \cdot d(a,c)^z + \left( \frac{2z+\varepsilon}{\varepsilon} \right)^{z-1} d(b,c)^z.$$

# B  Omitted Proofs for Center-Based Clustering

**Lemma B.1.** *Let $P \subset B_2^d$ be a set of points. Let $V$ be the set of all cost vectors of $P$ for $(k, z)$-clustering. Then there exists an $\varepsilon$-clustering net of size*

$$|\mathcal{N}(V, \|.\|_\infty, \varepsilon)| \le \exp(O(1) \cdot z \cdot k \cdot d \cdot \log(z\varepsilon^{-1})).$$

*Proof.* We start by proving the bound for $k = 1$. Suppose we are given a net $\mathcal{N}(B_2^d, \|.\|_2, \delta)$, for a $\delta$ to be determined later. Consider a candidate solution $\{s\}$ with cost vector $v^{\{s\}} \in V$. Let $s'$ be the point in $\in \mathcal{N}(B_2^d, \|.\|_2, \delta)$ of such that $\|s - s'\| \le \delta$, if $s'$ is not unique any one will be sufficient. Let $v^{\mathcal{S}'}$ be the cost vector of $\mathcal{S}'$. The number of distinct solutions $\mathcal{S}'$ are $|\mathcal{N}(B_2^d, \|.\|_2, \delta)| = \exp(O(1) \cdot d \cdot \log \delta^{-1})$ due to Lemma A.2.

What is left to show is that all solutions constructed in this way satisfy the guarantee of $\mathcal{N}(V, \|.\|_\infty, \delta)$, for an appropriately chosen $\delta$. We have for any $p \in P$ and any non-negative integer $z$ due to Lemma A.3

$$\begin{aligned} |\|p - s\|^z - \|p - s'\|^z| &\le \alpha \cdot \|p - s\|^z + \left( \frac{2z+\alpha}{\alpha} \right)^{z-1} \|s - s'\|^z \\ &\le \alpha \cdot \|p - s\|^z + (3z)^z \left( \frac{\delta}{\alpha} \right)^{z-1} \cdot \delta \end{aligned}$$

We set $\alpha = \frac{1}{2 \cdot 2^z} \varepsilon$ and $\delta = \alpha \cdot \frac{1}{2(3z)^z} \varepsilon = \frac{1}{4(6z)^z} \varepsilon^2$. Then the term above is upper bounded by at most $\varepsilon$ as $\|p - s\| \le 2$. Now since $|\|p - s\|^z - \|p - s'\|^z| \le \varepsilon$ for all $s \in B_2^d$ also implies $|\min_{s \in \mathcal{S}} \|p - s\|^z - \min_{s' \in \mathcal{S}'} \|p - s'\|^z| \le \varepsilon$, we have proven our desired approximation.

To conclude, observe that by our choice of $\delta$, the overall net $N$ has size at most $\exp(O(1) \cdot z \cdot d \cdot \log(z\varepsilon^{-1}))$.

To extend this proof to $k$-centers, observe that any solution consisting of $k$ centers can be obtained by selecting $k$ points from $B_2^d$, rather than one. This raises the net size of the single cluster case by a power of $k$. $\qquad \square$

We now show that Lemma B.1 combined with terminal embeddings yields the desired net.

**Lemma** (Equation 4 in Lemma 5.2). *Let $\mathcal{D}$ be a distribution over $B_2^d$ and let $P$ a set of $n$ points sampled from $\mathcal{D}$ and let $V$ be defined as in Theorem 2. Then*

$$|\mathcal{N}(V, \|.\|_\infty, \varepsilon)| \le \exp(O(1)z^3 \cdot k \cdot \varepsilon^{-2} \log n \cdot (\log(z) + \log(\varepsilon^{-1}))).$$

*Proof.* Let $f : \mathbb{R}^d \to \mathbb{R}^m$ be a terminal embedding, that is $f$ is such that $m \in O(z^2 \cdot \varepsilon^{-2} \log |P|)$[4] and for all $p \in P$ and $q \in \mathbb{R}^d$

$$\|p - q\|^z = (1 \pm \varepsilon)\|f(p) - f(q)\|^z,$$

as given by [62]. Therefore, for any candidate solution $\mathcal{S}$, we also have

$$\mathrm{cost}(p, \mathcal{S}) = (1 \pm 2\varepsilon)\mathrm{cost}(f(p), f(\mathcal{S})).$$

In other words, the set of cost vectors in the image of $f$ is the desired $O(\varepsilon)$-net for the true set of cost vectors. Hence an $\varepsilon$-net for the cost vectors induced by solutions in the image of $f$ is also an $O(\varepsilon)$-net for the set of cost vectors. We thus may apply Lemma B.1 for all cost vectors induced by solutions in the image of $f$. After rescaling $\varepsilon$ by constant factors, the overall net size is therefore $\exp(O(1)z^3 \cdot k \cdot \varepsilon^{-2} \log n \cdot (\log(z) + \log(\varepsilon^{-1})))$

$\square$

# C   Omitted Proofs for Subspace Clustering

In this section, we provide full proofs for Section 5 relative to subspace clustering.

We start with a few basic lemmas that will be useful in the calculations later.

We further require the following bounds that will prove useful in the calculations later.

**Lemma C.1.** *Let $a, b$ be numbers in $[0, 2]$ and let $\varepsilon > 0$. Suppose $a^2 = b^2 \pm \varepsilon \cdot b$. Then*

$$|a - b| \leq \varepsilon.$$

*Moreover, for any non-negative integer $z$, we have*

$$|a^z - b^z| \leq 2 \cdot (3z)^z \cdot \varepsilon.$$

*Proof.* For the first part of the lemma, we observe

$$|a^2 - b^2| = |a - b| \cdot (a + b) \leq \varepsilon \cdot b$$

which implies

$$|a - b| \leq \varepsilon.$$

For the second part, Lemma A.3 implies

$$|a^z - b^z| \leq \varepsilon \cdot \max(a, b)^z + \left(\frac{2z + \varepsilon}{\varepsilon}\right)^{z-1} \cdot |a - b|^z \leq \varepsilon \cdot 2^z + \left(\frac{3z + \varepsilon}{\varepsilon}\right)^{z-1} \cdot \varepsilon^z \leq 2(3z)^z \varepsilon. \quad \square$$

This lemma now immediately implies the following corollary by rescaling $\varepsilon$.

**Corollary C.2.** *Let $a, b$ be numbers in $[0, 2]$ and let $\varepsilon > 0$. Suppose $a^2 = b^2 \pm \frac{1}{4 \cdot (3z)^z} \max(\varepsilon \cdot b, \varepsilon^2)$. Then for any non-negative integer $z$, we have*

$$|a^z - b^z| \leq \varepsilon.$$

We now show that for any candidate subspace $U$ we can find a subspace representing it that is spanned by only a few vectors in $P$.

**Lemma** (Lemma 5.5). *Let $P \subseteq B_2^d$. For any orthogonal matrix $U \in \mathbb{R}^{j \times d}$, there exists $M \subseteq P$, with $|M| = O(j \cdot \varepsilon^{-2})$, such that*

$$\forall p \in P, \|U^T(I - \Pi_M)p\| \leq \varepsilon \cdot \|(I - \Pi_M)p\|. \tag{14}$$

*Proof.* Initially, let $M = \emptyset$. We add points to $M$ in rounds and denote by $M_t$ the set after $t$ rounds. Furthermore, let $\Pi_t$ be the projection matrix onto the subspace spanned by $M_t$ at round $t$. If there is a $p \in P$ in round $t$ such that

$$\|U^T(I - \Pi_t)p\| > \varepsilon\|(I - \Pi_t)p\| \tag{15}$$

---

[4]The dependency on $z$ is easily derived via a straightforward application of Lemma A.3.

then we let $M_{t+1} = M_t \cup \{p\}$. Our goal is to show that after $T \in O(j\varepsilon^{-2})$ many rounds, we have $\|U^T(I - \Pi_T)p\| \leq \varepsilon \cdot \|(I - \Pi_T)p\|$. We show this by proving inductively

$$\|U^T\Pi_t\|_F^2 \geq \varepsilon^2 \cdot t.$$

For the base case $t = 0$, this is trivially true. Thus suppose we add a point $p$ in iteration $t + 1$. Reformulating Equation 15, we have $\frac{\|U^T(I-\Pi_t)p\|}{\|(I-\Pi_t)p\|} > \varepsilon$. By the Pythagorean theorem, we therefore have

$$\|U^T\Pi_{t+1}\|_F^2 = \|U^T\Pi_t\|_F^2 + \frac{\|U^T(I - \Pi_t)p\|^2}{\|(I - \Pi_t)p\|^2} \geq \varepsilon^2 \cdot t + \varepsilon^2 \geq \varepsilon^2 \cdot (t+1).$$

Now since $\Pi_t$ is a projection and since $U$ has j orthonormal columns $j \geq \|U^T\|_F^2 \geq \|U^T\Pi_t\|_F^2$. If $T \geq \varepsilon^{-2}j$, we obtain $\|U^T\Pi_T\|_F^2 \geq j$. This implies that $U$ is contained in the space spanned by $M_T$. Conversely, $U$ must also be orthogonal to the kernel of $M_T$ that is $U(I - \Pi_T) = 0$. Therefore after at most $\varepsilon^{-2}j$ rounds, we have $\|U^T(I - \Pi_T)p\| \leq \varepsilon \cdot \|(I - \Pi_T)p\|$. $\qquad\square$

**Lemma** (Lemma 5.4). *Let $P \subset B_2^d$ be a set of points and let $z$ be a positive integer. Then there exists an $(\varepsilon, j)$-projective net of size*

$$|\mathcal{N}(V, \|.\|_\infty, \varepsilon)| \leq \exp(O(1) \cdot d \cdot j \cdot \log(j\varepsilon^{-1})).$$

*Proof.* Let $N$ be an $\varepsilon/j$-net of the Euclidean unit ball, i.e. $N = \mathcal{N}(B_2^d, \|.\|_2, \varepsilon/j)$ due to Lemma A.2. Let $\mathcal{N} = \otimes_{i=1}^j N$ be the set of $j-$subsets of of $N$. We claim that for every $S$, there exists an $S' \in \mathcal{N}$ such that

$$\|S^T p\|_2 = \|S'^T p\|_2 \pm \varepsilon.$$

Note that this implies the claim as $|\mathcal{N}| \in \left(\left(1 + \frac{2j}{\varepsilon}\right)^d\right)^j = \exp(O(1) \cdot d \cdot j \cdot \log(j\varepsilon^{-1}))$.

Define $S_i'^T$ to be the vector in $N$ closest to the $i$th row of $S^T$, i.e. $\|S_i^T - S_i'^T\|_2 \leq \varepsilon/j$. We have $\|S'^T - S\|_2 \leq \sum_{i=1}^j \|S'_i^T - S_i^T\|_2 \leq \varepsilon$. Therefore

$$
\begin{aligned}
\|S^T p\|_2 &= \|(S^T - S'^T)p + S'^T p\|_2 \\
&\leq \|(S^T - S'^T)p\|_2 + \|S'^T p\|_2 \\
&\leq \|S'^T p\|_2 + \|S^T - S'^T\|_2 \|p\|_2 \\
&\leq \|S'^T p\|_2 + \varepsilon.
\end{aligned}
$$

The bound $\|S^T p\|_2 \geq \|S'^T p\|_2 - \varepsilon$ is proven analogously. $\qquad\square$

We can now conclude with the proof of Equation 5 in Lemma 5.2.

**Lemma** ( Equation 5 in Lemma 5.2). *Let $\mathcal{D}$ be a distribution over $B_2^d$ and let $P$ a set of $n$ points sampled from $\mathcal{D}$ and let $V_{j,z}$ be defined as in Theorem 5.1. Then*

$$|\mathcal{N}(V_{j,z}, \|.\|_\infty, \varepsilon)| \leq \exp(O(1)(3z)^{z+2} \cdot k \cdot \varepsilon^{-2}(\log n + j \log(j\varepsilon^{-1})) \log \varepsilon^{-1}).$$

*Proof.* Let $\alpha, \beta > 0$ be sufficiently small parameters depending on $\varepsilon$ that will determined later. We first describe a construction for nets for a single subspace of rank at most $j$, before composing to $k$ subspaces.

We start by describing the composition of the nets. For every subset $M \subseteq P$, with $|M| \in O(j\alpha^{-2})$, we let $\Pi_M$ denote an orthogonal projection matrix of the span of $M$. Note that this implies $\mathbf{rank}(\Pi_M) = O(j\alpha^{-2})$. Further, let $N(\Pi_M) := \mathcal{N}(B_2^{\mathbf{rank}(\Pi_M)}, \|.\|_2, \beta)$ be a $(\beta, j)$-projective net of the point set $\cup_{p \in M}\{\Pi_M p\}$ of size at most $\exp(O(1) \cdot \mathbf{rank}(\Pi_M) \cdot \log(j\beta^{-1}))$ given by Lemma 5.4. Finally, let $N := \cup_M N(\Pi_M)$.

We consider an arbitrary orthogonal matrix $U \in \mathbb{R}^{j \times d}$. Denote by $M_U$ the subset of points and by $\Pi_U$ the projection matrix given by Lemma 5.5, using $\alpha$ as the precision variable. We claim that for every $U$, there exists an $U' \in N$ such that for all $p \in P$

$$\left| \left(\|\Pi_U p\|_2^2 - \|U'^T \Pi_U p\|_2^2 + \|(I - \Pi_U)p\|_2^2\right)^{z/2} - \|(I - UU^T)p\|^z \right| \in O(\alpha + \beta).$$

In other words, by enumerating over all $(\beta, j)$-projective nets, we obtain an $O(\alpha + \beta)$-subspace clustering net for $(1, j, z)$-clustering. The desired error of $\varepsilon$ then follows by choosing $\alpha$ and $\beta$ accordingly. For $U$, we construct $U'$ as follows. Let $D = \sqrt{\Pi_U}$, i.e. $DD^T = \Pi_U$. Further, let $V = U^T D$, notice that $V$ has at most $j$ rows that have at most unit norm. Hence, there exists a $U' \in N$ such that

$$\left| \|U\Pi_U p\|_2 - \|U'\Pi_U p\|_2 \right| \le \varepsilon$$

that is a $(\beta, j)$-projective net.

We then obtain

$$
\begin{aligned}
& \|\Pi_U p\|_2^2 - \|U'^T \Pi_U p\|_2^2 + \|(I - \Pi_U)p\|_2^2 \\
=\ & \|\Pi_U p\|_2^2 - \|U^T \Pi_U p\|_2^2 \pm \beta + \|(I - \Pi_U)p\|_2^2 \\
=\ & \|\Pi_U p\|_2^2 - \|UU^T \Pi_U p\|_2^2 \pm \beta + \|(I - \Pi_U)p\|_2^2 \\
=\ & \|(I - UU^T)\Pi_U p\|_2^2 + \|(I - \Pi_U)p\|_2^2 \pm \beta \\
(Eq.6)\quad =\ & \|(I - UU^T)p\|_2^2 \pm \beta - \|U^T(I - \Pi_U)p\|^2 - 2p^T \Pi_U UU^T (I - \Pi_U)^T p \\
(Lem.5.5)\quad =\ & \|(I - UU^T)p\|_2^2 \pm \alpha^2 \cdot \|(I - UU^T)p\|^2 \pm 2\alpha \cdot \|(I - UU^T)p\| \pm \beta
\end{aligned}
$$

Setting $\alpha^2 = \beta = \frac{1}{64(3z)^z}\varepsilon^2$, we then have due to Corollary C.2

$$\left| \left| \|\Pi_U p\|_2^2 - \|U'^T \Pi_U p\|_2^2 + \|(I - \Pi_U)p\|_2^2 \right|^z - \|(I - UU^T)p\|^z \right| \le \varepsilon. \qquad (16)$$

To extend this from a single $j$-dimensional subspace to a solution $\mathcal{U}$ given by the intersection of $k$ $j$-dimensional subspaces, we define cost vectors $v^{\mathcal{S}'}$ obtained from $\mathcal{N} = \otimes_{i=1}^k N$ as follows. For each $U \in \mathcal{U}$ let $U'$ be constructed as above and let $\mathcal{U}'$ be the union of the thus constructed $U'$. Then, with a slight abuse of notation, letting $\Pi_{U'}$ correspond to the subspace used to obtain $U'$, we define

$$v_p^{\mathcal{U}'} := \min_{U' \in \mathcal{U}'} \left| \|(I - \Pi_{U'})p\|^2 + \|\Pi_{U'}p\|^2 - \|U'\Pi_{U'}p\|^2 \right|^{z/2}.$$

Let $U$ be the subspace to which $p$ is assigned $\mathcal{U}$ and let $U'$ be the center in $\mathcal{U}'$ used to approximate $U$ and let $U^{*\prime} = \operatorname{argmin}_{U' \in \mathcal{U}'} \left| \|(I - \Pi'_U)p\|^2 + \|\Pi_{U'}p\|^2 - \|U'\Pi'_U p\|^2 \right|^{z/2}$ and let $U^* \in \mathcal{U}$ be the center approximated by $U^{*\prime}$. Then applying Equation 16, we have

$$
\begin{aligned}
& \|(I - UU^T)p\|^z \\
\le\ & \|(I - U^*U^{*T}p\|^z \\
\le\ & \left| \|(I - \Pi_{U^*})p\|^2 + \|\Pi_{U^*}p\|^2 - \|U*'\Pi_{U^*}p\|^2 \right|^{z/2} + \varepsilon \\
\le\ & \left| \|(I - \Pi_{U'})p\|^2 + \|\Pi_{U'}p\|^2 - \|U'\Pi_{U'}p\|^2 \right|^{z/2} + \varepsilon
\end{aligned}
$$

Thus, the cost vectors obtained from $\mathcal{N}$ are a $(k, j, z)$-clustering net, i.e.

$$\left| v_p^{\mathcal{S}'} - v_p^{\mathcal{S}} \right| := \left| \min_{s' \in \mathcal{S}'} \|\Pi_{s'}p - [s', 0]\|^z - \min_{s \in \mathcal{S}} \|p - s\|^z \right| \le \varepsilon.$$

What remains is to bound the size of the clustering net. Here we first observe that size of the clustering net is equal to $|\mathcal{N}| = |N|^k$. For $|N|$, we have $\binom{|P|}{O(\alpha^{-2}\log\alpha^{-1})} \le n^{O(j\alpha^{-2}\log\alpha^{-1})}$ many choices of $N(\Pi)$. In turn, the size of each $N(\Pi)$ is bounded by $(\beta/j)^{-O(j^2\alpha^{-2})}$ due to Lemma 5.4. Thus the overall size of $\mathcal{N}$ is

$$
\begin{aligned}
& \exp\left( k \cdot j \cdot O(\alpha^{-2}\log\alpha^{-1}(\log n + j\log\beta/j)) \right) \\
=\ & \exp(O(1)(3z)^{z+2} \cdot k \cdot j \cdot \varepsilon^{-2}(\log n + j\log(j\varepsilon^{-1}))\log\varepsilon^{-1})
\end{aligned}
$$

as desired. $\qquad\qquad \square$

## C.1 Proofs of Theorem 5.6 (Section 5.1)

The proof of the theorem is a straightforward application of Theorem 4.1 with the following Lemma

**Lemma C.3.** *Let $\mathcal{D}$ be a distribution over $B_2^d$, let $P$ a set of $n$ points sampled from $\mathcal{D}$, and let $V$ be defined as in Theorem 5.6. Then for any $\gamma > 0$*

$$Rad_n(V_{j,2}) \in O\left(\sqrt{\frac{kj}{n}\log^{3+\gamma}\left(\frac{n}{j}\right)}\right).$$

*Proof.* We use the following result due to Foster and Rakhlin [45].

**Theorem C.4** ($\ell_\infty$ contraction inequality (Theorem 1 by [45])). *Let $F \subseteq X \to \mathbb{R}^k$, and let $\phi : \mathbb{R}^k \to \mathbb{R}$ be L-Lipschitz with respect to the $\ell_\infty$ norm, i.e. $\|\phi(X) - \phi(X')\|_\infty \leq L \cdot \|X - X'\|_\infty$ for all $X, X' \in \mathbb{R}^k$. For any $\gamma > 0$, there exists a constant $C > 0$ such that if $|\phi_t(f(x))| \vee \|f(x)\|_\infty \leq \beta$, then*

$$Rad_n(\phi \circ F) \leq C \cdot L\sqrt{K} \cdot \max_i Rad_n(F|_i) \cdot \log^{3/2+\gamma}\left(\frac{\beta n}{\max_i R_n(F|_i)}\right).$$

We use this theorem as follows. Our functions are associated with candidate solutions $\mathcal{U}$, that is $\phi(f) = \min_{U \in \mathcal{U}} \|(I - UU^T)p\|_2^2$. In other words, $f$ maps a point $p$ to the $k$-dimensional vector, where $f_i(p) = \|(I - U_iU_i^T)p\|_2^2$ and $\phi$ selects the minimum value among all $\|I - U_iU_i^T)p\|_2^2$.

Thus, we require three more steps. First, we have to bound the Lipschitz constant of the minimum operator. Second, we have to give a bound on $\beta$. Third and last, we have to give a bound on the Rademacher complexity

$$Rad_n(V) = \frac{1}{n} \cdot \mathbb{E}_r \sup_U \sum_{p \in P} \|(I - UU^T)p\|_2^2 r_p. \tag{17}$$

The Lipschitz constant of the minimum operator with respect to the $\ell_\infty$ norm can be readily shown to be 1 as for any two vectors $x, y$ with $\min_i y_i = y_j$

$$\min_i x_i - \min_i y_i = \min_i x_i - y_j \leq x_j - y_j \leq |x_j - y_j| \leq \|x - y\|_\infty.$$

Since $U$ is an orthogonal matrices and $p \in B_2^d$, we have $\|(I - UU^T)p\|_2^2 \leq 1$ and thus $\beta$ is bounded by 1.

Thus, we only require a bound on Equation 17. For this, we use a result by [50]. Since the result is embedded in the proof of another result, we restate it here for the convenience of the reader.

**Lemma C.5** (Compare the proof Theorem 3 of [50]). *Let $P$ be an set of $n$ points in $B_2^d$ and let $\mathcal{U}$ be the set of all orthogonal matrices of rank at most $j$. For every $U \in \mathcal{U}$, define $f_U(p) = \|(I - UU^T)p\|_2^2$ and let $F$ be the set of all functions $f_U(p)$ Then.*

$$Rad_n(F) := \frac{1}{n} \cdot \mathbb{E}_r \sup_{U \in \mathcal{U}} \sum_{p \in P} \|(I - UU^T)p\|_2^2 \cdot r_p \in O\left(\sqrt{\frac{j}{n}}\right).$$

*Proof.* We have

$$Rad_n(F) = \mathbb{E}_r \sup_U \sum_{p \in P} \|(I - UU^T)p\|_2^2 r_p = \mathbb{E}_r \sum_{p \in P} \|p\|^2 r_p + \mathbb{E}_r \sup_U \sum_{p \in P} \|U^Tp\|_2^2 r_p.$$

We observe that the term $\mathbb{E}_r \sum_{p \in P} \|p\|^2 r_p$ is 0. Thus, we focus on the second term. We have

$$
\begin{aligned}
\mathbb{E}_r \sup_U \sum_{p \in P} \|U^T p\|_2^2 \cdot r_p &= \mathbb{E}_r \sup_U \sum_{p \in P} p^T U U^T p \cdot r_p = \mathbb{E}_r \sup_U \sum_{p \in P} trace(p^T U U^T p) \cdot r_p \\
&= \mathbb{E}_r \sup_U \sum_{p \in P} trace(U U^T p p^T) \cdot r_p \\
&= \mathbb{E}_r \sup_U trace\left( U U^T \sum_{p \in P} (r_p \cdot p p^T) \right) \\
&\leq \mathbb{E}_r \sup_U \|U\|_F \left\| \sum_{p \in P} r_p \cdot p p^T \right\|_F.
\end{aligned}
$$

We have $\|U\|_F \leq \sqrt{j}$, so we focus on $\left\| \sum_{p \in P} r_p \cdot p p^T \right\|_F$. Here, we have

$$
\begin{aligned}
\left\| \sum_{p \in P} r_p \cdot p p^T \right\|_F^2 &= trace\left( \left( \sum_{p \in P} r_p \cdot p p^T \right) \left( \sum_{p \in P} r_p \cdot p p^T \right) \right) \\
&= \sum_{p \in P} \sum_{q \in P} r_p \cdot r_q \cdot trace\left( p p^T q q^T \right) = \sum_{p \in P} \sum_{q \in P} r_p \cdot r_q \cdot (p^T q)^2.
\end{aligned}
$$

This implies

$$
\begin{aligned}
n \cdot Rad_n(F) &= \mathbb{E}_r \sup_U \sum_{p \in P} \|U^T p\|_2^2 r_p \leq \mathbb{E}_r \sup_U \|U\|_F \left\| \sum_{p \in P} r_p \cdot p p^T \right\|_F \\
&\leq \sqrt{j} \cdot \mathbb{E}_r \sqrt{\sum_{p \in P} \sum_{q \in P} r_p \cdot r_q \cdot (p^T q)^2} \\
\text{(Jensen's inequality)} \quad &\leq \sqrt{j} \cdot \sqrt{\mathbb{E}_r \sum_{p \in P} \sum_{q \in P} r_p \cdot r_q \cdot (p^T q)^2} \\
&= \sqrt{j} \cdot \sqrt{\sum_{p \in P} (p^T p)^2} \leq \sqrt{j} \cdot \sqrt{\sum_{p \in P} 1} = \sqrt{nj}.
\end{aligned}
$$

Solving the above for $Rad_n(F)$ concludes the proof. $\qquad \square$

We can now conclude the proof. Combining the bounds on $L$ and $\beta$ with Lemma C.5 and Theorem C.4, we have

$$
Rad_n(V_{j,2}) \in O\left( \sqrt{k} \cdot \sqrt{\frac{j}{n}} \cdot \log^{3+\gamma}(n) \right)
$$

as desired. $\qquad \square$

## C.2 Lower Bound

Finally, we also show that the bound given in Theorem 5.6 is optimal, up to polylog factors.

**Theorem** (5.7). *There exists a distribution $\mathcal{D}$ supported on $B_2^d$ such that $\mathcal{E}(V_{j,2}) \in \Omega\left( \sqrt{\frac{kj}{n}} \right)$.*

*Proof.* We first describe the hard instance distribution $\mathcal{D}$. We assume that we are given $d = 2kj$ dimensions. Let $e_i$ be the standard unit vector along dimension $i$ with $i \in \{1, \ldots d\}$. Let $p, \varepsilon \in [0, 1]$ be a parameters, where $\varepsilon$ is sufficiently small. We set the densities for a point $q$ as follows.

$$
\mathbb{P}[q] = \begin{cases} p & \text{if } q = e_i, i \in \{1, \ldots, k \cdot j\} \\ p - \varepsilon \cdot p & \text{if } q = e_i, i \in \{kj + 1, \ldots, d\} \\ 0 & \text{otherwise} \end{cases} \tag{18}
$$

We choose $p$ such that integral over densities is 1, i.e. $kj \cdot p + kj \cdot (p - \varepsilon p) = 1$. It is straightforward to verify that for $\varepsilon$ sufficiently small, $p \in (\frac{1}{kj}, \frac{2}{kj})$. We denote the points $\{e_1, \ldots e_{kj}\}$ by $G$ for "good" and the points $\{e_{kj+1}, \ldots e_d\}$ by $B$ for "bad".

We now characterize the properties of the optimal solution as well as suboptimal solutions.

**Lemma C.6.** *Let $\mathcal{D}$ be the distribution described above in Equation (18). Then for any optimal solution $\mathcal{U} = \{U_1, \ldots U_k\}$, we have $e_i \in U_t$ for $i \in \{1, \ldots, kj\}$ and some $t$ and $OPT = kj \cdot p \cdot (1 - \varepsilon)$.*

*Proof.* We transform the instance into a $d \times d$ diagonal matrix $D$ where $D_{i,i} = \sqrt{\mathbb{P}[e_i]}$. So $D$ is a $d \times d$ diagonal matrix with diagonal entries equal to $\sqrt{p}$ for the first $k \cdot j$ elements and $\sqrt{p - \varepsilon \cdot p}$ for elements from $k \cdot j + 1$ to $d$. Now consider any partition of the points into clusters $C_t$ with the corresponding subspace $U_t$ for ($t \in \{1, \ldots, k\}$ ). The optimal solution for $U_t$ is simply the right singular vector of the submatrix of $D$ corresponding to points in $C_t$, which by the construction of $D$ is the $j$ points with the largest weight. This means that each cluster can remove at most $\sum_{i=1}^{j} 1 = j$ from the cost, so $k$ clusters can remove at most $\sum_{i=1}^{k} j$ from the cost. This imples that the cost of the clustering is lower bounded by $\sum_{i=1}^{d} D_{i,i}^2 - \sum_{i=1}^{kj} D_{i,i}^2 = \sum_{i=kj+1}^{d} D_{i,i}^2$. Conversely, the solution $\mathcal{U}$ has exactly this cost, which implies that it must be optimal. $\square$

Using Lemma C.6, we now have to, given $n$ independent samples from $\mathcal{D}$. Control the probability that the sample $P$ will (falsely) put a higher weight on some of the points in $B$ than the points in $G$. Let $B_{ex}$ denote the set of misclassified points in $B$ and let $P_{\text{OPT}}$ denote the optimum computed on the sample $P$. We have

$$\mathbb{E}[\text{cost}(\mathcal{D}, P_{\text{OPT}})] = kj \cdot p \cdot (1 - \varepsilon) + p \cdot \varepsilon \cdot |B_{ex}|.$$

and hence an expected excess risk bound of

$$\mathbb{E}[\text{cost}(\mathcal{D}, P_{\text{OPT}})] - \text{OPT} = p \cdot \varepsilon \cdot \mathbb{E}[B_{ex}].$$

By linearity of expectation, we have $\mathbb{E}[|B_{ex}|] = kj \cdot \mathbb{P}[e_{kj+1} \in B_{ex}]$. Thus, $\mathbb{E}[\text{cost}(\mathcal{D}, P_{\text{OPT}})] - \text{OPT} \in \Theta(1)\varepsilon \cdot \mathbb{P}[e_{kj+1} \in B_{ex}]$. Define $G_{low}$ to be the set of points from $G$ that are have an empirical density of at most $p$. Let $\widehat{e_{kj+1}}$ denote the empirical density of $e_{kj+1}$. We now claim that

$$\begin{aligned}
\mathbb{P}[e_{kj} \in B_{ex}] &\geq \mathbb{P}[\widehat{e_{kj+1}} > p \wedge e_{kj+1} \in B_{ex}] \\
&= \mathbb{P}[e_{kj+1} \in B_{ex} | \widehat{e_{kj+1}} > p] \cdot \mathbb{P}[\widehat{e_{kj+1}} > p] \geq 1/2 \cdot \mathbb{P}[\widehat{e_{kj+1}} > p]
\end{aligned}$$

The first inequality follows because we are considering a subset of the possible events, the second inequality follows because the number of points with an empirical estimated density greater than $p$ is negatively correlated with the empirical density $\widehat{e_{kj+1}}$ of the point $e_{kj}$. Specifically, conditioned on $\widehat{e_{kj+1}} > p$, the mean and median density of any point $e_i \in G$ is at most $\frac{1}{n} \cdot p(n - p \cdot n) = p \cdot (1 - p) < p$. Thus, the (marginal) mean and median density of any other point is below $p$ and therefore the probability that $e_{kj+1}$ will be in $B_{ex}$ is at least $1/2$.

Thus, what remains to be shown is a bound on $\mathbb{P}[e_{kj} > p]$. Here, we use the tightness of the Chernoff bound (see Lemma 4 of [48]).

**Lemma C.7** (Tightness of the Chernoff Bound). *Let $X$ be the average of $n$ independent, $0/1$ random variables. For any $\varepsilon \in (0, 1/2]$ and $\mu \in (0, 1/2]$, assuming $\varepsilon^2 \mu n \geq 3$ if each random variable is 1 with probability at least $\mu$, then*

$$\mathbb{P}[X > (1 + \varepsilon)p] > \exp(-9\varepsilon^2 \mu n).$$

Thus, sampling $n$ elements, we have

$$\begin{aligned}
\mathbb{P}[e_{kj} > p] = \mathbb{P}\left[e_{kj} > \left(1 + \frac{\varepsilon}{1 - \varepsilon}\right) \cdot (1 - \varepsilon) \cdot p\right] \\
> \exp\left(-9\frac{\varepsilon^2}{(1 - \varepsilon)^2}(1 - \varepsilon)pn\right) \in \Omega(1) \exp\left(-\frac{\varepsilon^2}{kj}n\right).
\end{aligned}$$

If we require $\mathbb{E}[\text{cost}(\mathcal{D}, P_{\text{OPT}})] - \text{OPT} = \varepsilon \cdot c$ for a sufficiently small absolute constant $c$, we also require $\mathbb{P}[e_{kj} > p] = c'$ and hence $\sqrt{\frac{kj}{n}} \leq \varepsilon \cdot c''$ for a sufficiently small absolute constants $c'$ and $c''$. Letting $\varepsilon \to 0$ then shows that the excess risk can asymptotically decrease no faster than $\Omega\left(\sqrt{\frac{kj}{n}}\right)$. $\qquad\square$

# D  Details for the Experiments (Section 6)

## D.1  Description of datasets

Mushroom comprises of 112 categorical features of the appearance of mushrooms with class labels corresponding to poisonous or edible. MNIST contains 28x28 pixel images of handwritten digits. Skin_Nonskin are RGB values given as 3 numerical features used to predict if a pixel is skin or not. Lastly, Covtype consists of a mix of categorical and numerical features used to predict seven different cover types of forests. In the main body, we focus on Covtype because of its large number of points.

## D.2  Description of algorithms

**Center based clustering**    For each experiment, we use an expectation maximization (EM) type algorithm. Given a solution $\mathcal{S}$, we first assign every point to its closest center and subsequently, we recompute the center. For the case $z = 2$, we do this analytically and in this case the EM algorithm is more commonly known as Llyod's method [59]. For the cases, $z \in \{1, 3, 4\}$, the new center is obtained via gradient descent. The initial centers are chosen via $D^z$ sampling, i.e. sampling centers proportionate to the $z$th power of the distance between a point and its closest center (for $z = 2$ this is the $k$-means++ algorithm by [6]).

We wrote all of the code using Python 3 and utilized the Pytorch library for implementations using gradient descent. Specifically, we employed the AdamW optimizer to find the closest center with a learning rate set to $0.01$. All experiments were conducted on a machine equipped with a single NVIDIA RTX 2080 GPU.

**Subspace Clustering**    For subspace clustering, we consider $j \in \{1, 2, 5\}$ to demonstrate the effects of the subspace dimension on convergence rate, taking computational expenses into consideration. Since there are no known tractable algorithms for these problems with guarantees, we initialize a solution $\mathcal{U} = \{U_1, \ldots, U_k\}$ by sampling $k$ orthogonal matrices of rank $j$, where the subspace for each matrix is determined via the volume sampling technique [35]. Subsequently, we run the EM algorithm. As before, the expectation step consists of finding the closest subspace for every point. For $z = 2$, the maximization step consists of finding the $j$ principal component vectors of the data matrix induced by each cluster. For the other values of $z$, it is NP-hard even approximate the maximization step [24], so we use gradient descent to find a local optimum. Due to the fact that Skin_nonskin only has 3 features, we only evaluate the excess risk for $j \in \{1, 2\}$. Due to a large computational dependency on dimension, we do not evaluate subspaces on the MNIST dataset.

## D.3  Experimental results

In this section, we provide plots of the excess risk and the found parameters of the best-fit lines for each of the datasets.

Table 2: Best fit lines on Covtype and Mushroom (left to right)

| z | c | $q_1$ | $q_2$ | z | c | $q_1$ | $q_2$ |
|---|---|---|---|---|---|---|---|
| 1 | $3 \cdot 10^{-2}$ | 0.44 | 0.54 | 1 | $1 \cdot 10^{-1}$ | 0.48 | 0.51 |
| 2 | $4 \cdot 10^{-3}$ | 0.42 | 0.52 | 2 | $8 \cdot 10^{-2}$ | 0.48 | 0.51 |
| 3 | $6 \cdot 10^{-4}$ | 0.44 | 0.51 | 3 | $4 \cdot 10^{-2}$ | 0.49 | 0.50 |
| 4 | $1 \cdot 10^{-4}$ | 0.44 | 0.51 | 4 | $3 \cdot 10^{-2}$ | 0.49 | 0.50 |

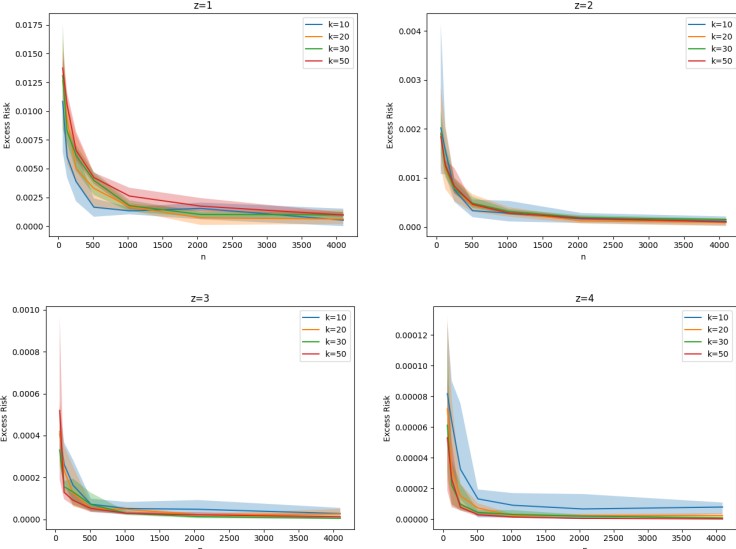

Figure 2: Excess risk for center-based clustering on the Covertype dataset. The shaded areas indicate the maximal and minimal deviation for the respective sample sizes.

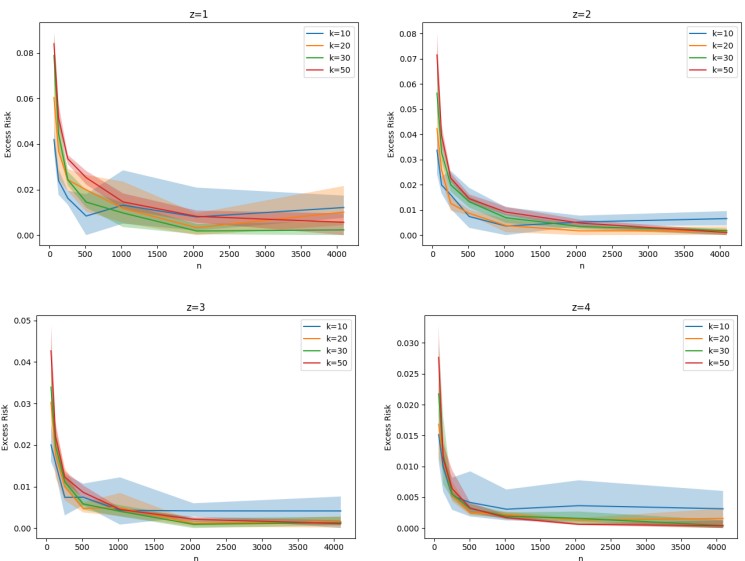

Figure 3: Excess risk for center-based clustering on the Mushroom dataset. The shaded areas indicate the maximal and minimal deviation for the respective sample sizes.

Table 3: Best fit lines on Skin_NonSkin and MNIST (left to right)

| z | $c$ | $q_1$ | $q_2$ | z | $c$ | $q_1$ | $q_2$ |
|---|---|---|---|---|---|---|---|
| 1 | $2 \cdot 10^{-2}$ | 0.49 | 0.50 | 1 | $1 \cdot 10^{-1}$ | 0.49 | 0.51 |
| 2 | $3 \cdot 10^{-3}$ | 0.47 | 0.52 | 3 | $5 \cdot 10^{-2}$ | 0.50 | 0.50 |
| 3 | $8 \cdot 10^{-4}$ | 0.46 | 0.53 | 4 | $3 \cdot 10^{-2}$ | 0.50 | 0.50 |
| 4 | $2 \cdot 10^{-4}$ | 0.46 | 0.53 | 2 | $8 \cdot 10^{-02}$ | 0.50 | 0.50 |

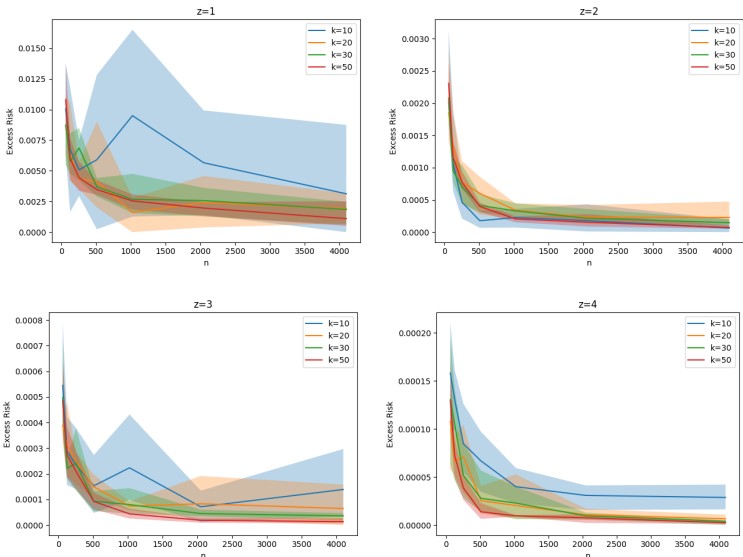

Figure 4: Excess risk for center-based clustering on the Skin_Nonskin dataset. The shaded areas indicate the maximal and minimal deviation for the respective sample sizes.

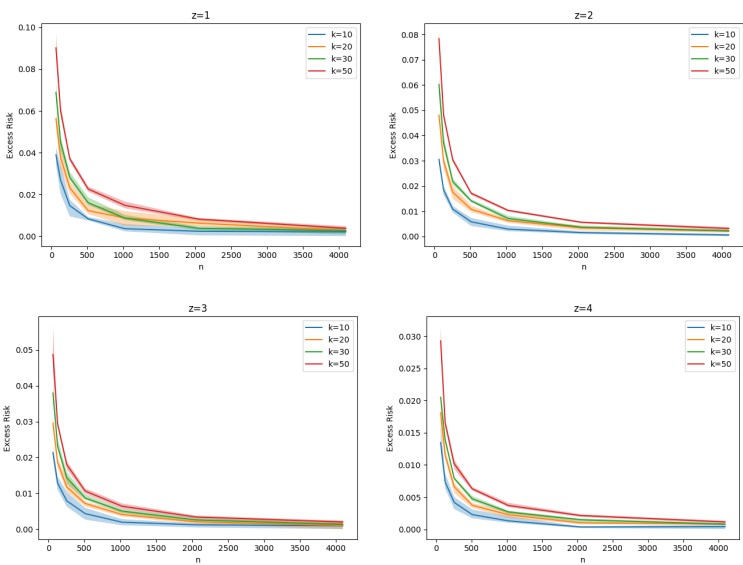

Figure 5: Excess risk for center-based clustering on the MNIST dataset. The shaded areas indicate the maximal and minimal deviation for the respective sample sizes.

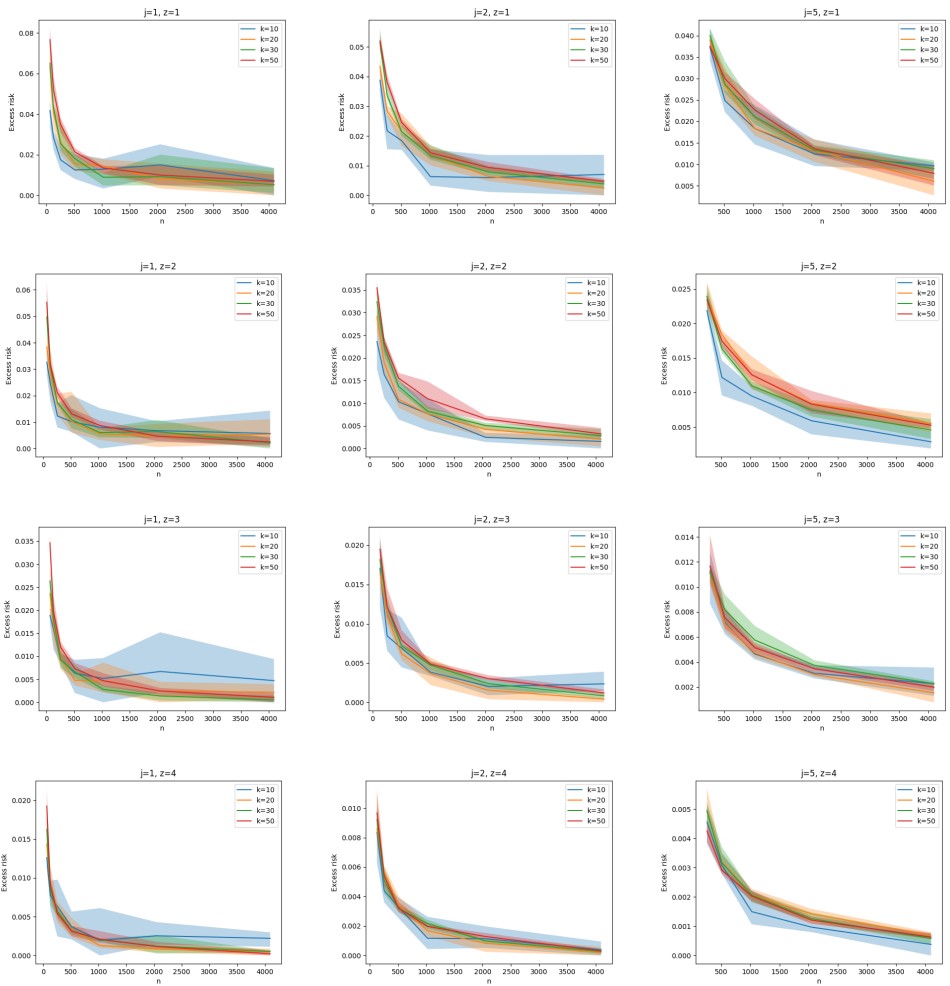

Figure 6: Excess risk for subspace clustering on the Mushroom dataset. The shaded areas indicate min/max values

Table 4: Best fit line for subspace clustering on Covtype and Mushroom (left to right)

| j | z | $c$ | $q_1$ | $q_2$ | j | z | $c$ | $q_1$ | $q_2$ |
|---|---|---|---|---|---|---|---|---|---|
| 1 | 1 | $0.1$ | 0.45 | 0.54 | 1 | 1 | $1 \cdot 10^{-1}$ | 0.48 | 0.51 |
| 1 | 2 | $2 \cdot 10^{-2}$ | 0.48 | 0.51 | 1 | 2 | $1 \cdot 10^{-1}$ | 0.48 | 0.51 |
| 1 | 3 | $3 \cdot 10^{-4}$ | 0.46 | 0.53 | 1 | 5 | $1 \cdot 10^{-1}$ | 0.49 | 0.49 |
| 1 | 4 | $4 \cdot 10^{-5}$ | 0.46 | 0.52 | 2 | 1 | $7 \cdot 10^{-2}$ | 0.48 | 0.51 |
| 2 | 1 | $8 \cdot 10^{-2}$ | 0.48 | 0.51 | 2 | 2 | $6 \cdot 10^{-2}$ | 0.50 | 0.49 |
| 2 | 2 | $2 \cdot 10^{-3}$ | 0.47 | 0.51 | 2 | 5 | $6 \cdot 10^{-2}$ | 0.49 | 0.48 |
| 2 | 3 | $4 \cdot 10^{-5}$ | 0.46 | 0.53 | 3 | 1 | $4 \cdot 10^{-2}$ | 0.49 | 0.50 |
| 2 | 4 | $2 \cdot 10^{-6}$ | 0.46 | 0.52 | 3 | 2 | $3 \cdot 10^{-2}$ | 0.49 | 0.50 |
| 5 | 1 | $8 \cdot 10^{-3}$ | 0.48 | 0.51 | 3 | 5 | $3 \cdot 10^{-2}$ | 0.49 | 0.49 |
| 5 | 2 | $5 \cdot 10^{-5}$ | 0.46 | 0.53 | 4 | 1 | $2 \cdot 10^{-2}$ | 0.49 | 0.50 |
| 5 | 3 | $4 \cdot 10^{-7}$ | 0.47 | 0.52 | 4 | 2 | $2 \cdot 10^{-2}$ | 0.49 | 0.50 |
| 5 | 4 | $3 \cdot 10^{-9}$ | 0.47 | 0.51 | 4 | 5 | $1 \cdot 10^{-2}$ | 0.48 | 0.50 |

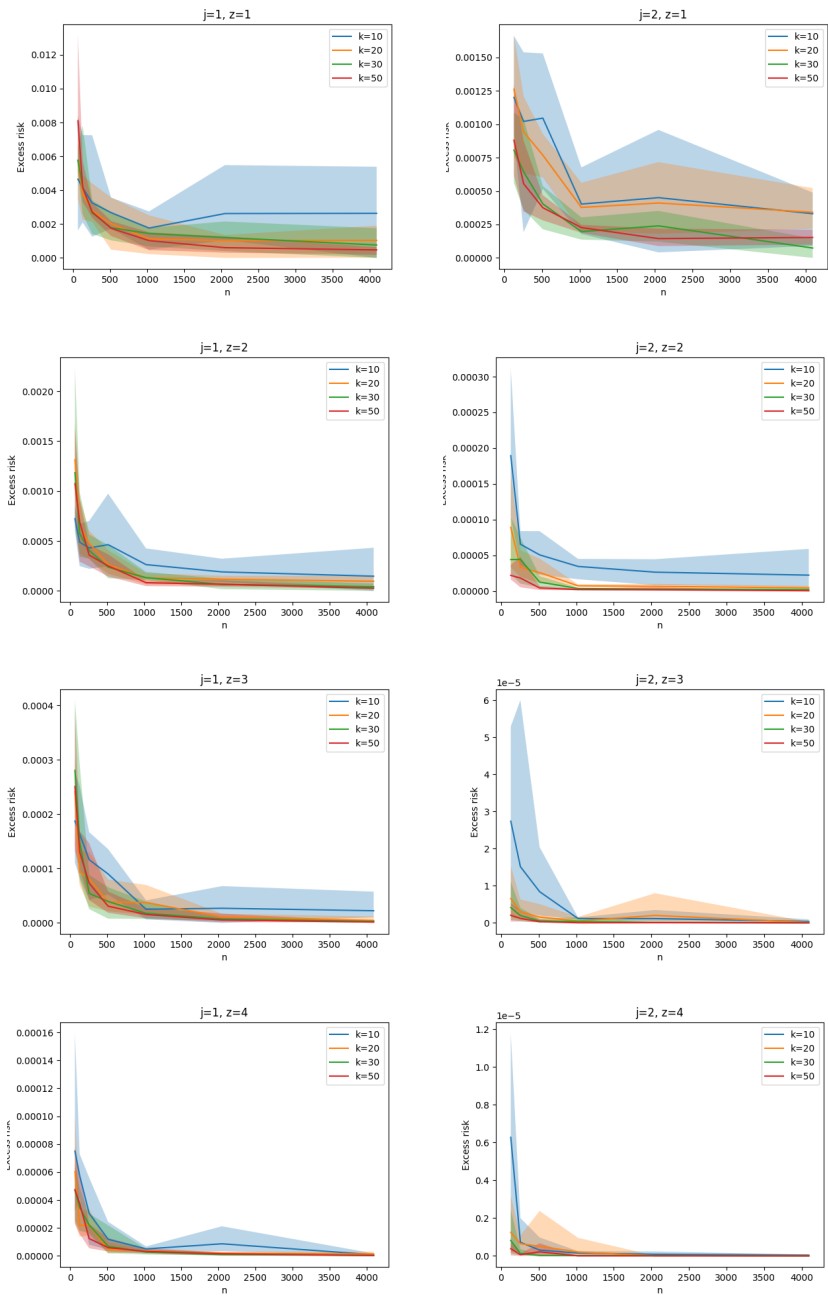

Figure 7: Excess risk for subspace clustering on the Skin_Nonskin dataset. The shaded areas indicate min/max values

Table 5: Best fit line for subspace clustering on Skin-Nonskin

| j | z | $c$ | $q_1$ | $q_2$ |
|---|---|---|---|---|
| 1 | 1 | $1 \cdot 10^{-2}$ | 0.48 | 0.50 |
| 1 | 2 | $3 \cdot 10^{-3}$ | 0.45 | 0.53 |
| 2 | 1 | $2 \cdot 10^{-3}$ | 0.46 | 0.53 |
| 2 | 2 | $2 \cdot 10^{-4}$ | 0.46 | 0.53 |
| 3 | 1 | $4 \cdot 10^{-4}$ | 0.46 | 0.53 |
| 3 | 2 | $2 \cdot 10^{-5}$ | 0.46 | 0.53 |
| 4 | 1 | $9 \cdot 10^{-5}$ | 0.46 | 0.53 |
| 4 | 2 | $3 \cdot 10^{-6}$ | 0.46 | 0.53 |

