# OpenReview forum: "On Generalization Bounds for Projective Clustering"
_NeurIPS.cc/2023/Conference — NeurIPS 2023 poster_

### Official Review · Reviewer_26Fm · 2023-07-02

**Soundness:** 4 excellent
**Presentation:** 3 good
**Contribution:** 3 good
**Rating:** 7
**Confidence:** 3

**Summary:**

The authors study the generalization bounds for two clustering problems: center-based clustering and subspace (projective) clustering. The authors argue that the problems reduce to bounding  the Gaussian complexity of the set of cost functions over all possible solutions. To achieve this, they apply the union bound on the telescoping sum over a nested sequence of solutions sets that are increasingly more accurate. To find a set of solutions at a specific level of accuracy, the authors use the \emph{terminal embedding} for the center-based clustering, and they propose a new dimensionality reduction method for the projective clustering.

**Strengths:**

- The authors provide the first provable bound for projective clustering. For a specific case with the squared cost, they show that the previous bound by Fefferman, Mitter, and Narayanan (2016) is optimal.
- I find the chaining technique to be quite interesting, and it could be applied to other learning problems.
- The authors did a decent job of giving a high level idea of their proofs
- The experimental result on real datasets agree with the theoretical results.


**Weaknesses:**

My only concerns are regarding the applications and experiments. Since this is mainly theoretical paper, these should not be major concerns.

- The problem of projective clustering should be better motivated in the introduction. What are some real applications of projective clustering? What are the common choices of $j$?
- The experiment are performed with $j\in \{1,2,5\}$, with only $j=1$ presented in the main paper. In my opinion, there should be a bit of discussion on how the experimental results conform (or deviate) from theory as $j$ increases.



**Questions:**

- As the authors mentioned in D.2, the projective clustering is serverly limited in computational aspect. Has there been any attempt to solve this issue? The authors probably should mention this limitation somewhere in the paper as well.

**Limitations:**

The authors have mentioned several open problems related to this work, but I think the most important problem is whether the projective clustering rate of $\tilde{O}(\sqrt{kj^2/n})$ is optimal for $z\not= 2$.

---

> ### Author Rebuttal · Authors · 2023-08-02
>
> We thank the reviewer for the comments.
>
> Regarding the questions and weaknesses:
>
> To the best of our knowledge, the most commonly used choice of $j$ are small constants. For example [1] never use $j$ larger than 4. The reason why we focussed on $j=1$ in the main body is that there already exists a phase transition in terms of the computational complexity between the normal $k$-median and $k$-means problems and using lines as centers, while still admitting more positive results than other subspace clustering problems [2,3,4]. In addition, the problem is often considered in computational geometry as it can be interpreted as finding the ($k$) closest cylinders. But we are open to putting more attention on other values of $j$.
>
> In terms of tractability: the $(k,1,z)$ problem is inapproximable up to any factor, even in 2 dimensions, for any choice of $z$ [5]. Similarly, the $(1,j,z)$ problem is APX hard [6,7] unless $z=2$ in which case the problem is variously known as PCA or low rank approximation. To run an EM-like algorithm for $(k,j,z)$ clustering, as we did in the experiments, we require a very accurate (read $(1+\epsilon)$ approximation). The currently best known algorithms for $z=1$ and $n$ points run in time $O(\exp((j\cdot\epsilon^{-1})^{O(z)})$ [6,8]. Unfortunately, running these algorithms even for small values of $j$ is still very impractical. Thus we relied on heuristic EM-like methods which is also what [1] suggests.
>
> [1] René Vidal: Subspace Clustering. IEEE Signal Process. Mag. 28(2): 52-68 (2011)
>
> [2] Dan Feldman, Amos Fiat, Micha Sharir: Coresets forWeighted Facilities and Their Applications. FOCS 2006
>
> [3] Dan Feldman, Amos Fiat, Micha Sharir, Danny Segev: Bi-criteria linear-time approximations for generalized k-mean/median/center. SCG 2007
>
> [4] Pankaj K. Agarwal, Cecilia Magdalena Procopiuc, Kasturi R. Varadarajan: Approximation Algorithms for a k-Line Center. Algorithmica 42(3-4): 221-230 (2005)
>
> [5] V. S. Anil Kumar, Sunil Arya, H. Ramesh: Hardness of Set Cover with Intersection 1. ICALP 2000
>
> [6] Kenneth L. Clarkson, David P. Woodruff: Input Sparsity and Hardness for Robust Subspace Approximation. FOCS 2015
>
> [7] Amit Deshpande, Madhur Tulsiani, Nisheeth K. Vishnoi: Algorithms and Hardness for Subspace Approximation. SODA 2011
>
> [8] Dan Feldman, Michael Langberg: A unified framework for approximating and clustering data. STOC 2011

---

> > ### Comment · Reviewer_26Fm · 2023-08-17
> > **Response**
> >
> > I thank the authors for addressing my concerns and for the references. As the proved convergence rates are quite general and distribution-free, I think the paper provides significant contribution.

---

### Official Review · Reviewer_iH2U · 2023-07-08

**Soundness:** 3 good
**Presentation:** 4 excellent
**Contribution:** 3 good
**Rating:** 7
**Confidence:** 3

**Summary:**

This paper investigates generalization bounds for center based and subspace clustering, providing upper bounds on excess risk for $(k,z)$ and $(k,j,z)$ clustering and lower bound for $(k,j,z)$ clustering for special case of $z=2$.

The bounds for $(k,j,z)$ clustering are novel and lower bounds helps establish optimality of previously known bounds.

**Strengths:**

The paper is well written and proof sketch is easy to follow.

This work provides improvements over existing work in extending $(k,z)$ clustering bounds. Getting chaining type analysis to work in this setting is non-trivial.

The techniques used for proving upper bounds for subspace clustering are novel and using multiple dimensionality reductions for analysis is indeed interesting.

The lower bound construction is clean. Overall, I like the results in this paper.

I have not verified all the proofs in the appendix but the proof sketch and flow seems right to me.

**Weaknesses:**

The bounds are interesting in certain parameter regimes, specifically when $j$ and $z$ are constants. Limitations in current approach to extend beyond this is not discussed.

**Questions:**

Q1: Could this approach also help obtain improvements for when additional structural assumptions are imposed on $\mathcal{D}$?

Q2: What changes in bound for Lemma 4.4 when points do not lie in low-dimensional space?

**Limitations:**

Please refer weakness and questions. Work is theoretical in nature, negative societal impact is not apparent.

---

> ### Author Rebuttal · Authors · 2023-08-02
>
> We acknowledge the reviewer's comments.
>
> Regarding the limitations in $j$ and $z$, for $z\rightarrow\infty$, no generalization bounds are possible, at least in this problem setup. The main issue is that problems with $z\rightarrow\infty$ are too sensitive to outliers and even regions with an extremely low density are important. An exponential depedency on $z$ is likely necessary, as for $z'\in \Omega(\log n)$, where $n$ is the sample size, the cost of a $(k,j,z')$ clustering and a $(k,j,\infty)$ clustering is very close.
> For large values of $j$, there are no limitations in our analysis, barring perhaps a quadratic dependecy for $z\neq 2$. The only reason why we mentioned that $j$ is a constant because this is typically assumed in practice [7]. The main limitations are only in terms of finding a tractable algorithm that solves the problem. For example, for $(k,1,z)$ clustering, the problem is inapproximable even in the Euclidean plane [1]. Exponential time algorithms that solve these problems exist [2,3], but are mainly of theoretical interest.
>
> Q1: It may be possible.  For $k$-means, previous work has achieved a learning rate of $O(1/n)$ under certain assumptions, see [4,5]. It may be possible to weaken these assumptions and/or extend these ideas to k-median. It is, for example, a very interesting open problem to show that assuming ORSS-stability [6], a learning rate of $O(k/n)$ is possible. Doing so will likely require a different approach than what we are currently doing.
> For $(k,j,z)$ clustering, we are not aware of any prior results. It is likely that some assumptions exist such that this possible.
>
> Q2: This lemma itself does not change. But using our ensemble of dimension reductions, we obtain the bounds for Lemma 4.5, which are independent of $d$. We only use lemma 4.4. once we know that we can map the points to some low-dimensional space. If the dependency on $d$ can be eliminated entirely, one would bypass the necessity for dimension reduction and directly achieve an optimal learning rate for $(k,j,z)$ clustering.
>
> [1] V. S. Anil Kumar, Sunil Arya, H. Ramesh: Hardness of Set Cover with Intersection 1. ICALP 2000
>
> [2] Kenneth L. Clarkson, David P. Woodruff: Input Sparsity and Hardness for Robust Subspace Approximation.FOCS 2015
>
> [3] Dan Feldman, Michael Langberg: A unified framework for approximating and clustering data. STOC 2011
>
> [4] C. Levrard. Nonasymptotic bounds for vector quantization in hilbert spaces. The Annals of Statistics 2015
>
> [5] Shaojie Li, Yong Liu: Sharper Generalization Bounds for Clustering. ICML 2021
>
> [6] Rafail Ostrovsky, Yuval Rabani, Leonard J. Schulman, Chaitanya Swamy: The effectiveness of lloyd-type methods for the k-means problem. J. ACM 59(6): 28:1-28:22 (2012)
>
> [7] René Vidal: Subspace Clustering. IEEE Signal Process. Mag. 28(2): 52-68 (2011)

---

> > ### Comment · Reviewer_iH2U · 2023-08-16
> > **Official comment**
> >
> > Thanks for the response and indulging the questions. I believe the paper makes significant contribution so I maintain my score.

---

### Official Review · Reviewer_ZgJL · 2023-07-09

**Soundness:** 3 good
**Presentation:** 3 good
**Contribution:** 3 good
**Rating:** 5
**Confidence:** 4

**Summary:**

This paper presents several generalization bounds for clustering objectives such as
k-median and subspace clustering. When the centers are points or constant dimensional subspaces, the upper bounds are optimal up to logarithmic terms. For projective clustering, this work gives a lower bound showing that the results obtained by [34] are nearly optimal. A key technique was using an ensemble of dimension reduction methods with guarantees.

**Strengths:**

This paper has the following contributions:
For center-based objectives, this work shows a convergence rate, which matches the known optimal bounds for k-means, and extends it to other important objectives such as k-median. For subspace clustering with j-dimensional subspaces, this work also shows a convergence rate. For the specific case of projective clustering, which generalizes k-means, a converge rate is provided.

**Weaknesses:**

1.Insufficient research on related work.
This work is not the first random projection clustering work, such as papers [1,2]. These papers are not cited in this paper. The superiority of this work cannot be verified. Please compare them from the theoretical analysis, experimental results, algorithm complexity, etc.

[1]Yin R, Liu Y, Wang W, et al. Randomized Sketches for Clustering: Fast and Optimal Kernel $ k $-Means[J]. Advances in Neural Information Processing Systems, 2022, 35: 6424-6436.

[2]Yin R, Liu Y, Wang W, et al. Scalable Kernel $ k $-Means with Randomized Sketching: From Theory to Algorithm[J]. IEEE Transactions on Knowledge and Data Engineering, 2022.

2.The experiments is not enough. There are many related works in this field. If this paper can be compared with related work through experiments and the performance of the work can be analyzed in detail, it would be better.

3.The presentation of references is not standardized, such as:

[20] P. Chou. The distortion of vector quantizers trained on n vectors decreases to the optimum as Op(1/n). In Proceedings of 1994 IEEE International Symposium on Information Theory, pages 457–, 1994. doi: 10.1109/ISIT.1994.395072.

[28] V. Cohen-Addad, D. Saulpic, and C. Schwiegelshohn. Improved coresets and sublinear algorithms for power means in euclidean spaces. Advances in Neural Information Processing Systems, 34, 2021.

[50] Y. Liu, S. Liao, S. Jiang, L. Ding, H. Lin, and W. Wang. Fast cross-validation for kernel-based algorithms. IEEE Transactions on Pattern Analysis and Machine Intelligence, PP:1–1, 01 2019. doi: 10.1109/TPAMI.2019.2892371.

4.The organization and presentation of this paper can be further improved.


**Questions:**

See “Weaknesses”.

---

> ### Author Rebuttal · Authors · 2023-08-02
>
> We will add the desired aforementioned references. We can also standardize the references.
>
> However, we seriously question the validity of the criticism that we did not compare "theoretical analysis, experimental results, algorithm complexity, etc". In words of the reviewer, "This work is not the first random projection clustering work" and the reviewer seems to imply that the two proposed references are the first to do so.
> 1. The proposed references focus on kernel k-means. k-means is not the primary subject of this study, having been solved optimally in previous work and our techniques go beyond what is possible for k-means, and indeed what is being done in those two papers. Our considered objective functions and settings are different.
> 2. The reviewer seems to think that random projections are a core part of our work. This is false. Nowhere do we use random projections for $(k,j,z)$ clustering in any way and we specifically made a point to discuss shortcomings of all existing dimension reduction methods.
> 3. Random projections for clustering have been used as early as 2010 by [BZD], and subsequently studied in [CEMMP,MMR,BBCGS], all of which have appeared before the two papers mentioned by the reviewer and, it must be said, both failed to cite any of these earlier works. With the exception of [BZD], which we admittedly forgot and will rectify, these papers *are* in fact cited by us.
>
> Thus we strongly believe that the criticism that we did not compare "theoretical analysis, experimental results, algorithm complexity, etc" to the references suggested by the reviewer or that we did not do due diligence with the related work has no basis whatsoever.
>
> As for the organization, we would appreciate if the reviewer could provide details.
>
> [BZD] Christos Boutsidis, Anastasios Zouzias, Petros Drineas: Random Projections for $k$-means Clustering. NIPS 2010
>
> [CEMMP] Michael B. Cohen, Sam Elder, Cameron Musco, Christopher Musco, Madalina Persu: Dimensionality Reduction for k-Means Clustering and Low Rank Approximation. STOC 2015
>
> [MMR] Konstantin Makarychev, Yury Makarychev, Ilya P. Razenshteyn: Performance of Johnson-Lindenstrauss transform for k-means and k-medians clustering. STOC 2019
>
> [BBCGS] Luca Becchetti, Marc Bury, Vincent Cohen-Addad, Fabrizio Grandoni, Chris Schwiegelshohn: Oblivious dimension reduction for k-means: beyond subspaces and the Johnson-Lindenstrauss lemma. STOC 2019

---

### Official Review · Reviewer_zjTi · 2023-07-17

**Soundness:** 3 good
**Presentation:** 4 excellent
**Contribution:** 3 good
**Rating:** 7
**Confidence:** 3

**Summary:**

The authors study two clustering problems from the perspective of generalization:
-standard center-based clustering objectives such as k-means, k-median and more generally the different norms associated with the objective
-projective subspace clustering: where the goal is to find a k subspaces such that if you project the points there, you minimise a natural distance objective.

The main question addressed for both of these problems is:
If we are given a sample set of n data points drawn independently from a fixed (unknown) distribution, and we perform clustering on those n points, how fast will the solution on the sample converge to the optimal clustering on the fixed (unknown) distribution?



**Strengths:**

+presentation and literature review is done in a careful manner
+results for both clustering objectives are novel and interesting
+generalization bounds almost tight

**Weaknesses:**

-A lot of the tools needed by the authors seem to have been used in prior works too. Having said that, there are some proposed new techniques for dimension reduction which, though tailored to the problem at hand, seems interesting.

**Questions:**

N/A

---

### Official Review · Reviewer_hJ3D · 2023-07-26

**Soundness:** 3 good
**Presentation:** 3 good
**Contribution:** 3 good
**Rating:** 7
**Confidence:** 2

**Summary:**

This paper studies generalization bounds for clustering problems, including ceter-based clustering and subspace clustering. For center-based clustering, they recover the optimal bound (up to log term) of $\widetilde{O}(\sqrt{k/n})$; the technique can be extended to k-median also. For subspace clustering, the derive the first bound $\widetilde{O}(\sqrt{kj^2/n})$ for the generalized $(k, j, z)$-clustering objectives, which had been established only for the scenario $z = 2$; for the special case  $z=2$, they further refine the bound to $\widetilde{O}(\sqrt{kj/n})$ that meets the current upper-bound in the literature, and prove that this is tight. Experiments are given.

**Strengths:**

The paper advances the knowledge on generalization bound for general subspace clustering, which is the main merit of this paper. The authors also prove the tightness for the case $z=2$.

Given the intricate relevance between the clustering problem with coreset construction, dimension reduction and so on, the paper did a good job on discussing the relevant work and laying out their proof sketch, while pointing out the challenges for generalizing the results (from  $z=2$ as in the paper's reference [34]) to the case $(k, j, z)$. It is then clear that the chaining technique, which has resulted in tight bounds for coreset clustering and especially $(k, j, 2)$ clustering, is not compatible with existing (generic) dimension reduction techniques. From there, the authors design an ad-hoc dimension reduction technique for subspace clustering to overcome the obstacle. Several insights and techniques in the paper are thus new and of independent interests, while such arguments seem restrictively applicable to only clustering case.


**Weaknesses:**

While the paper is good in terms of technicality and novelty, it can be made more interesting if the authors can discuss better on use cases and importance of $z > 2$. In fact, the case $z=2$, which would correspond to the $\ell_2$  loss seems much more prevalent and important in practice. The current few-line discussion in the paper is a bit succinct and also does not point to any reference.



**Questions:**

N.A.

**Limitations:**

Limitations are addressed. There is no negative societal impact of the work.

---

> ### Author Rebuttal · Authors · 2023-08-03
>
> We thank the reviewer for the comments.
>
> The loss functions for different values of $z$ are often considered in theoretical papers (for instance [26,35]). The most important variant for large values of $z$ is undoubtedly $z\rightarrow \infty$, as it generalizes problems such as the minimum enclosing ball and $k$-center. Nevertheless, for the learning setting, large values of $z$ render the problem infeasible.
>
> As a more feasible alternative with provable guarantees, one can interpolate between and $\ell_2$ loss and $\ell_{\infty}$ by considering higher higher values of $z$, which has been done, for example, in [28]. Moreover, as observed by [*] (who we forgot to add to the references but should be included), learning bounds for $k$-means in $\ell_1$ space can be obtained by studying $(k,4)$ clustering in $\ell_2^2$ space.
>
> We will add these references and discussion in the potential final version.
>
> [26] V. Cohen-Addad, D. Saulpic, and C. Schwiegelshohn. A new coreset framework for clustering. STOC 2021
>
> [28] V. Cohen-Addad, D. Saulpic, and C. Schwiegelshohn. Improved coresets and sublinear algorithms for power means in euclidean spaces. Advances in Neural Information Processing Systems, 34, 2021.
>
> [35] D. Feldman and M. Langberg. A unified framework for approximating and clustering data. STOC 2011
>
> [*] Lingxiao Huang, Nisheeth K. Vishnoi: Coresets for clustering in Euclidean spaces: importance sampling is nearly optimal. STOC 2020

---

### Decision · Program_Chairs · 2023-09-21

**Decision:**

Accept (poster)

**Comment:**

This paper provides the first upper and lower generalization bounds for subspace clustering. The reviewers had a variety of small comments and are overall in favor, I am glad to recommend acceptance.

Minor comment: "converge rate" in the abstract should be "convergence rate".